

# Deciphering seasonal effects of triggering and preparatory precipitation for improved shallow landslide prediction using generalized additive mixed models

Stefan Steger[1], Mateo Moreno[1,2], Alice Crespi[1], Peter James Zellner[1], Stefano Luigi Gariano[3], Maria Teresa Brunetti[3], Massimo Melillo[3], Silvia Peruccacci[3], Francesco Marra[4], Robin Kohrs[1,5], Jason Goetz[5], Volkmar Mair[6], Massimiliano Pittore[1]

[1]Institute for Earth Observation, Eurac Research, Bolzano-Bozen, Italy
[2]University of Twente, Faculty of Geo-Information Science and Earth Observation (ITC), Enschede, The Netherlands
[3]CNR IRPI, Perugia, Italy
[4] Institute of Atmospheric Sciences and Climate, National Research Council (CNR-ISAC), Bologna, Italy
[5]Department of Geography, Friedrich Schiller University Jena, Germany
[6]Office for Geology and Building Materials Testing, Autonomous Province of Bolzano-South Tyrol, Cardano, Italy

*Correspondence to*: Stefan Steger (stefan.steger@eurac.edu)

**Abstract.**

The increasing availability of long-term observational data can lead to the development of innovative modelling approaches to determine landslide triggering conditions at regional scale, opening new avenues for landslide prediction and early warning. This research blends the strengths of existing approaches with the capabilities of generalized additive mixed models (GAMMs)
to develop an interpretable approach that identifies seasonally dynamic precipitation conditions for shallow landslides. The model builds upon a 21-year record of landslides in South Tyrol (Italy) and separates precipitation that induced landslides from precipitation that did not. The model accounts for effects acting at four temporal scales: short-term "triggering" precipitation, medium-term "preparatory" precipitation, seasonal effects and across-year data variability. It provides relative landslide probability scores that were used to establish seasonally dynamic thresholds with optimal performance in terms of hit and false
alarm rates, as well as additional thresholds related to user-defined performance scores. The GAMM shows a high predictive performance and indicates that more precipitation is required to induce a landslide in summer than in winter/spring, which can presumably be attributed mainly to vegetation and temperature effects. The discussion illustrates why the quality of input data, study design and model transparency are crucial for landslide prediction using advanced data-driven techniques.

## 1 Introduction

Landslides are potentially hazardous phenomena that can occur under a variety of environmental conditions. In mountainous and hillslope environments, they are amongst the most influential landscape-forming geomorphic processes while





simultaneously posing a risk to people, assets and infrastructure (Crozier, 1989; Glade et al., 2005). Each year, considerable economic losses and fatalities are caused by landslides across the globe and ongoing environmental changes are expected to alter future landslide hazard and associated risk (Slaymaker et al., 2009; Petley, 2012; Gariano and Guzzetti, 2016; Alvioli et

al., 2018; Froude and Petley, 2018; Haque et al., 2019; Lin et al., 2022; Maraun et al., 2022; Ozturk et al., 2022).

In this context, the development of reliable decision support tools and landslide early warning systems are of critical value to mitigate or reduce future harms. Literature depicts that landslide early warning systems have gained increasing attention over time (Piciullo et al., 2018; Guzzetti et al., 2020). At the same time, however, it is known that the effectiveness of subsequent decisions and risk reduction measures heavily depends on the reliability of the underlying prediction rules that describe the

associations between the occurrence of slope instabilities and their causes (Basher, 2006; Segoni et al., 2018b; Steger et al., 2021). In most cases, the timing of landslide occurrence can primarily be explained by an interplay of preparatory and triggering dynamic conditions, such as pre-moistened soil experiencing a heavy precipitation event (Crozier, 1989). In Italy, for instance, rainfall can be considered the main dynamic factor for explaining the timing of shallow landslide occurrence (Brunetti et al., 2010).

For regional scale assessments, statistical approaches are widely used to create an empirical relation between a landslide catalogue (i.e., information on past landslide occurrence) and associated rainfall measurements. For this purpose, a regression model is commonly used to derive empirical rainfall thresholds, i.e., a landslide is likely to be initiated if the respective rainfall conditions are exceeded. Widely used examples are the cumulative event rainfall–duration (*ED*) threshold or its widely used derivative, the intensity-duration (*ID*) threshold. These thresholds are visualized as lines within a log-scaled scatterplot (e.g.,

cumulated rainfall vs rainfall duration) of points, each representing a rainfall condition that induced one or more landslides. *ED* and *ID* thresholds may represent a low bound of rainfall amount below which past landslides have rarely been triggered – e.g., 5% non-exceedance probability – and future landslides are more likely expected with higher rainfall conditions (Caine, 1980; Guzzetti et al., 2008; Brunetti et al., 2010; Peruccacci et al., 2012; Piciullo et al., 2017; Guzzetti et al., 2022). Although such binary thresholds may be preferred for decision-making purposes, they do not ideally represent the continuous nature of

the underlying phenomena, i.e., the fact that the likelihood of landslide occurrence can change gradually with changing precipitation conditions. Empirical *ED* or *ID* thresholds are dependent on a variety of decisions (Segoni et al., 2018a), such as the type of rainfall data used (e.g., ground-based measurements or radar estimates) and their temporal and spatial resolutions, the specific criteria used to define the rainfall conditions (either triggering or non-triggering landslides), the strategy applied to link a specific landslide location with nearby rainfall measurement locations, and the method adopted to estimate the

threshold parameters. In particular, the separation of independent rainfall events is usually based on a criterion that refers to dry (or low rainfall) hiatuses between the events. Literature shows that fixed time periods (e.g., 24 hours without rainfall) and variable time periods, also depending on seasonality, have been proposed (Guzzetti et al., 2008; Brunetti et al., 2010; Melillo et al., 2015). Dedicated software able to objectively reconstruct rainfall events and to calculate rainfall thresholds for different non-exceedance probabilities has facilitated the automatic derivation of empirical thresholds (Melillo et al., 2018). In summary,

most empirical rainfall thresholds only consider rainfall conditions associated with known slope instabilities, neglecting those





that did not trigger landslides. Despite promising results from numerical and synthetic experiments (e.g., Peres and Cancelliere, 2021), it remains challenging to distinguish precipitation conditions that triggered landslides from those that did not trigger landslides at regional scale. In fact, lack of landslide information does not necessarily mean that landslides have not occurred in an area (Gariano et al., 2015; Steger et al., 2021).

Binary classification algorithms, such as logistic regression, are less used in this field, despite their potential to elaborate differences between rainfall that initiated landslides and rainfall that did not cause slope failure (Glade et al., 2000; Jakob and Weatherly, 2003; Frattini et al., 2009; Peres and Cancelliere, 2014; Giannecchini et al., 2016; Postance et al., 2018). This may be due to a non-trivial sampling of the representative non-landslide data (i.e., rainfall events not inducing landslides), associated data uncertainties and higher data demands (e.g., collection of landslide absence data).

Many empirical studies dealing with critical landslide rainfall conditions did not perform a quantitative result validation (Gariano et al., 2015; Segoni et al., 2018a). Approaches focusing on landslide-observations only (e.g., conventional ED thresholds) frequently compare the respective results with independent landslide datasets, thus providing insights into the portion of correctly "predicted" landslide events (Gariano et al., 2015; Piciullo et al., 2017). The trade-off between correctly classified landslide observations and correctly classified non-landslide observations is more straightforward to validate using

binary algorithms. In this context, confusion matrices or Receiver Operating Characteristic (ROC) curves can be used to evaluate the overall ability of the model to separate rainfall conditions with landslides from those without. ROC curves are primarily used to assess overall model performance for continuous outcomes (e.g., probability of landslides), but proved useful to derive optimized probability cutpoints (Jakob and Weatherly, 2003; Peres and Cancelliere, 2014; Gariano et al., 2015; Piciullo et al., 2017; Postance et al., 2018).

Although most empirical studies put the spotlight on short-term triggering conditions, the effects of preparatory hydrological factors and medium-term antecedent precipitation conditions on the occurrence of landslides were also explored (Bogaard and Greco, 2018). Usually, a variable or function representing medium-term precipitation before the triggering event is used as a proxy for subsurface wetness immediately prior to slope failure (Crozier, 1999; Glade et al., 2000; Mirus et al., 2018; Monsieurs et al., 2019; Leonarduzzi and Molnar, 2020; Rosi et al., 2021). For instance, early works of Glade et al. (2000) and

Chleborad (2000) built upon the idea that rainfall conditions responsible for landslide initiation can be described by combining a variable that represents the short-term precipitation immediately before slope failure with a variable that relates to the medium-term antecedent wetness conditions. In these cases, short-term triggering precipitation was represented by the total rainfall amount measured on the day of the event (Glade et al., 2000) or by the cumulative precipitation observed during a number of days prior to the event (e.g., 3 days in Chleborad, 2000). Antecedent precipitation was described by a time-

dependent decay coefficient (Crozier, 1999; Glade et al. 2000) or by cumulative precipitation in a fixed 15-day time window prior to the three-day "triggering" precipitation (Chleborad, 2000). Several works still rely on the idea of explicitly combining variables representing triggering precipitation and antecedent preparatory precipitation or soil moisture conditions to derive critical thresholds for landslide occurrence (Scheevel et al., 2017; Mirus et al., 2018; Postance et al., 2018; Rosi et al., 2021).



Rosi et al. (2021) observed that 3D thresholds based on intensity, duration and 7 to 30 days cumulative rainfall can considerably
reduce the false alarms in comparison to classical *ID* thresholds.

Novel data-driven approaches can go beyond a predetermined parametric equation and allow accounting for different
interacting effects on landslide occurrence in a flexible way (Steger et al., 2021; Distefano et al., 2022; Tehrani et al., 2022).
For instance, the consideration of seasonal interactions may allow to describe how the effect of a specific precipitation amount
on landslide occurrence may vary between seasons, since associated temperature and vegetation effects are known to induce
changes in interception, evaporation, plant water uptake and transpiration. This in turn affects slope hydrology and therefore
also slope stability (Sidle and Ochiai, 2006; Norris, 2008; Gonzalez-Ollauri and Mickovski, 2017; Schmaltz et al., 2018).
Seasons can also be associated with different types of precipitation, e.g., convective vs frontal events (Peruccacci et al., 2012)
and in some areas also snow melt plays a considerable role on slope stability (Krøgli et al., 2018; Schmaltz et al., 2018).
Comparable amounts of short-term precipitation may induce different quantities of landslides due to seasonal effects and
associated antecedent rainfall conditions (Luna and Korup, 2022).

In summary, no standard procedure exists for the identification of precipitation responsible for landslide occurrence and
different approaches have their advantages and drawbacks (Segoni et al., 2018a). This research aims to exploit the possibilities
offered by novel Generalized Additive Mixed Models (GAMMs) and the strengths of a wide range of existing approaches to
develop a flexible modelling procedure to identify critical, season-dependent rainfall conditions for shallow landslide
triggering. The proposed approach aims to create a data-driven model that:

- explicitly accounts for precipitation that caused landslides and precipitation that did not cause landslides;
- considers both short-term "triggering" precipitation (*T*) and medium-term "preparatory" precipitation (*P*);
- includes season-specific variations of the effect of *T* and *P* on landslide occurrence;
- considers potential inconsistencies in landslide reporting across the years (i.e., reporting biases)
- produces continuous outcomes (i.e., probability scores) that can be converted into objective optimal or user-defined thresholds based on ROC metrics;
- allows a thorough spatial and temporal cross-validation of the results.

The analysis builds upon a 21-year (2000-2020) record of earth and debris slides that occurred in South Tyrol (Northern Italy)
and high-resolution gridded daily precipitation data (Steger et al., 2021; Crespi et al., 2021).

**2 Study area**

The approach was developed for the mountainous Italian province of South Tyrol, northern Italy (Fig. 1a). The landslide-prone
area covers 7400 km² and is characterized by varying environmental conditions that determine landslide occurrence. Landslide



predisposition is primarily controlled by the general high relief energy, terrain morphology (mean slope 27°), variations in hillslope materials and vegetation cover (more details in Steger et al., 2021).

South Tyrol is located in the Southern Alps, which are at the intersection area for different air masses: humid influences from the Atlantic, dry air masses from the continental east and warm contribution from the Mediterranean. The geographical location together with the mountainous topography leads to a strong climate variability and several small-scale phenomena, including

thermal inversions and cool air pools, warm and dry Föhn events, orographic enhancement of precipitation and sheltered dry regions. The warmest conditions occur in July (13 °C as areal average) and the lowest temperatures in January (-4 °C) with a mean annual cycle showing the greatest warming between April and May and the largest cooling in the transition from October to November (Adler et al., 2015; Crespi et al., 2021). The wettest season is summer with about 120 mm/month (areal average) which also corresponds to the main convective period, while the lowest precipitation amounts are registered in winter,

especially in February (30 mm/month as areal average). Precipitation amounts are still remarkably high in autumn (~ 80 mm/month) when intense and persistent precipitation events are generally brought by low-pressure systems from a still warm Mediterranean Sea (Adler et al., 2015; Miglietta and Davolio, 2022). At annual scale, the wettest areas are located along the northern border of Pusteria Valley (east) and close to Trentino (south-east) where annual totals exceed 1500 mm, while drier conditions characterize the inner valleys. In particular, the Venosta valley, i.e., the west-to-east oriented valley in the western

part of the province, is one of the driest inner-Alpine spots where the rain-shadow effect determined by the surrounding high mountain ranges turns into less than 600 mm per year (Fig. 1b).

Heavy precipitation can be considered the primary triggering factor for shallow landslides in the area. However, snow melting and human activities were also reported to induce shallow slope failures (Tasser et al., 2003; Stingl and Mair, 2005; Piacentini et al., 2012; Schlögel et al., 2020; Steger et al., 2021). For South Tyrol, previous research focused on the definition of debris

flow triggering conditions and on the evaluation of uncertainties related to the use of different rainfall data sources (Nikolopoulos et al., 2015; Marra et al., 2016; Destro et al., 2017; Nikolopoulos et al., 2017; Martinengo et al., 2021). So far, no regional-scale research has been conducted to define critical precipitation conditions for shallow slide-type movements in South Tyrol.

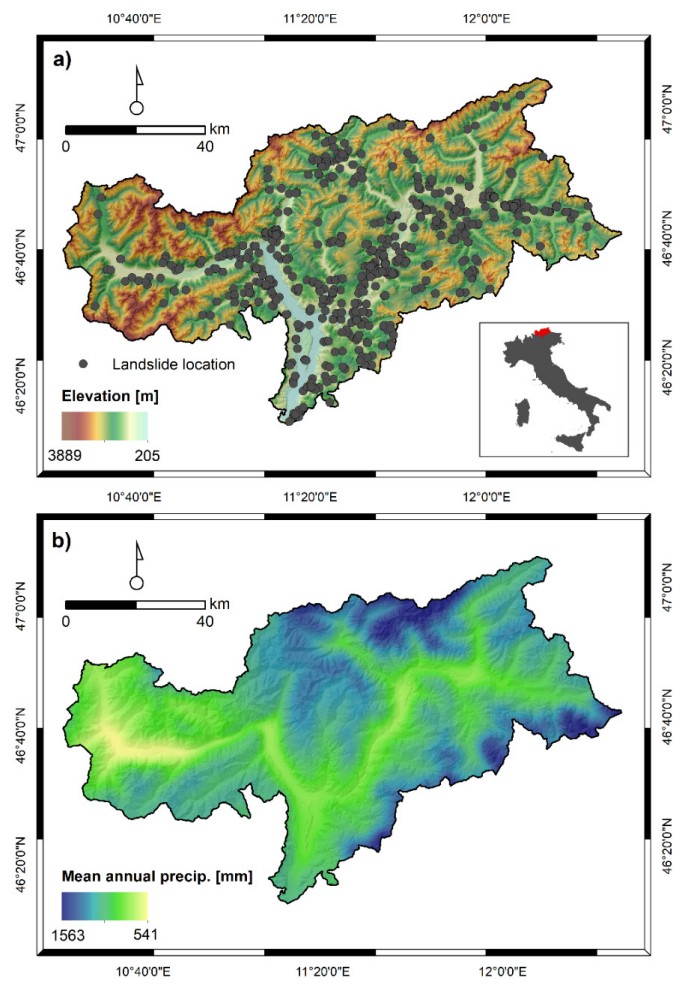


**Figure 1: Position of the study area within Italy (insert within a), elevation, and landslide scarp locations of the earth and debris slides (n=581) used for modelling (a), mean annual precipitation rates for the period 2000 to 2020 (b).**

### 3 Data

#### 3.1 Landslide data

The landslide data used for this study is based on the South Tyrolean version of the Italian landslide database, called IFFI (*Inventario dei Fenomeni Franosi in Italia*; Trigila et al., 2010). Background information on the South Tyrolean data set and its specificities, such as an inherent spatially variable completeness, are described in Steger et al. (2021). The South Tyrolean



IFFI data collection can be considered fairly systematic since the year 2000. However, temporal data heterogeneities across the years may still exist due to changes in the personnel responsible for compiling and digitizing the landslide information in different subregions. The comprehensive documentation of each registered landslides is associated with a variety of attributes that were further utilized to filter suitable entries. In summary, the used attributes relate to the occurrence date of the landslides, the movement-type, the material-type and the assigned movement-cause. Point-data associated to the initiation zone of each landslide was accessed from the IdroGeo platform (https://idrogeo.isprambiente.it/app/). From January 2000 to the end of 2020, a total of 11420 points related to different movement types (incl. pre-2000 events) were registered for South Tyrol.

Additionally, IFFI-independent records of landslides that occurred in South Tyrol between July 2003 and November 2019 were considered. These 47 data entries (further referred to as "IRPI landslide records") belong to a national catalogue which was compiled by gathering information from various sources, such as online newspapers, technical reports and blog entries (Brunetti et al., 2015; Peruccacci et al., 2017). Prior to the analysis, both the IFFI and the IRPI landslide records were subjected to a comprehensive data filtering process as described in Section 4.1.

## 3.2 Gridded precipitation data

The gridded fields of daily precipitation for South Tyrol were extracted from the 1980-2018 dataset produced by Crespi et al. (2021). The dataset exhibits a spatial resolution of 250 m and was extended until August 2021 to cover all landslide observations. The precipitation fields were obtained by interpolating the rain-gauge daily records from a quality-checked and homogenized archive including around 80 station sites from the weather station network of South Tyrol. The interpolation onto the target 250-m grid is based on a three-step scheme:

1. the 1981-2010 monthly precipitation averages of the stations were interpolated by means of a weighted linear regression with elevation whose coefficient was estimated monthly and at each grid-point from surrounding station values; station weights accounted for their distance from the point and the level of similarity in terms of elevation, slope steepness and slope orientation.

2. the daily station anomalies, i.e., the ratio of daily precipitation and the climatology of the corresponding month, were interpolated via an inverse distance weighting scheme where weights depended on horizontal and vertical distances from the target grid-point.

3. the final fields of daily precipitation are obtained by multiplying the gridded daily anomalies times the gridded climatologies for the corresponding month. Based on the common standard adopted by the weather data providers in Trentino-South Tyrol, the daily precipitation fields refer to the total precipitation occurring from 8:00 UTC of the previous day to 8:00 UTC of the observation day.

The orographic information at 250 m was computed by using the European Digital Elevation Model (EU-DEM) v1.1 from Copernicus (http://land.copernicus.eu/pan-european/satellite-derived-products/eu-dem/eu-dem-v1.1/view). Further details on the interpolation procedure are provided in Crespi et al. (2021).





The leave-one-out validation against station observations showed no systematic biases and a mean absolute error, as average over all months and station sites, of 1.1 mm. The uncertainty of interpolated precipitation decreases with increasing elevation along with the decrease of rain-gauge density (only 4% of stations in South Tyrol are in the elevation range 2000-2500 m). This effect, however, was considered less influential in this study, since the available landslide data systematically refers to lower altitude areas close to infrastructure (Steger et al., 2021). Another potential source of uncertainty, which was not

addressed in this procedure, is the wind-induced under-catch, especially at higher elevations during snowfalls, which could lead to a general underestimation of actual precipitation (Sevruk et al., 2009). Before landslide modelling, the 1.1 mm mean absolute error was used as an additional data filter criterion to differentiate between "rainy" days that experienced precipitation shortly before or at the sampling day (> 1.1 mm) from "dry" days (≤ 1.1 mm) that were not considered in the analyses (details in Section 4.2).

### 205    3.3 Additional environmental data for validation

Cross-validation (CV) based on a leave-one-factor-out data partitioning (Section 4.5) was based on several spatial environmental variables (Fig. 2). This data was obtained from the open Geodata platform of South Tyrol (Geokatalog, 2021). Model transferability amongst lithological units was tested based on a 1:500000 overview lithological map (Fig. 2a; Geologische Übersichtskarte Südtirol). A 30 m x 30 m bilinearly resampled LiDAR-DTM was used to derive slope angle

classes (Fig. 2b) and elevation classes (Fig. 2c). For these two variables, quartiles were calculated based on the number of landslide observations. Thus, each of the four classes is represented by an equal amount of landslide observations (i.e., 25%). The area was additionally subdivided into four large spatial blocks by merging neighbouring catchment areas to test the spatial transferability across these large subregions (Fig. 2d). To evaluate differences of model transferability across the land cover units, we took advantage of the associated IFFI attributes that refer to the land cover observed at the time of landslide initiation

(classes: forest, agricultural land and other/na).

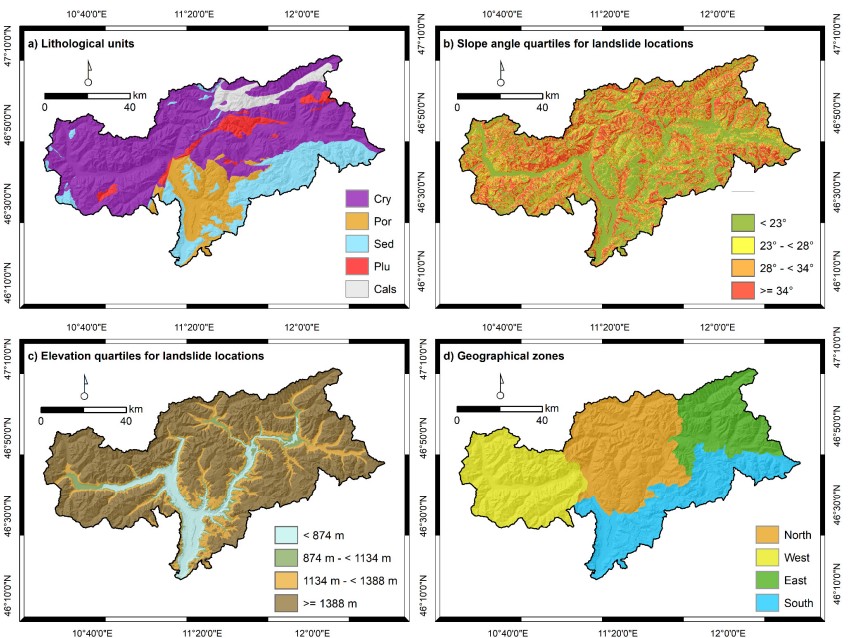

**Figure 2 Spatial environmental maps used for leave-one-factor-out CV. The lithological map (a) shows the classes Crystalline basement (Cry), Porphyry (Por), Sedimentary rocks (Sed), Plutonite (Plu) and Calcschists with ophiolites (Cals). The quartile classification for slope angle (b) and elevation (c) is based on the final landslide presence sample (i.e., each class contains 25% of landslides). The geographical zonation (d) is based on the intersection of catchments.**

## 4 Methods

The general workflow of this study is shown in Figure 3, while details are depicted in the Sections 4.1 to 4.5. In general, the binary response variable (landslide presences vs absences) was compiled by combining filtered information on precipitation-induced earth and debris slides with a sample of pre-landslide dates at these locations (Section 4.1). Each observation of the resulting binary variable was then used to extract a variety of pre-observation cumulative precipitation variables on the basis of daily gridded precipitation data (Section 3.2). In analogy to Chleborad (2000), two different types of precipitation variables were extracted for each observation: six candidates for representing potential triggering precipitation $T$ (0 to 5 days prior the observation) and 30 candidates for representing medium-term antecedent cumulative precipitation $P$ (1 to 30 days prior $T$) (Section 4.2). A parsimonious and interpretable model was created by identifying the best-performing $T$-$P$ combination, i.e., one $T$ and one $P$ variable. For this purpose, a two-parameter ($T_{days}$, $P_{days}$) grid-search combined with CV was used to iteratively change and test the predictive performance of 165 $T$-$P$ combinations (Section 4.3). A binomial Generalized Additive Mixed Model (GAMM) enabled to model the binary response as a (non-linear) function of the main effects $T$, $P$ and a seasonal cyclic effect (day-of-the-year; $DOY$). Also, potential interactions between the main effects ($T$, $P$, $DOY$) were considered via additional


tensor product interaction terms. Furthermore, two random effects were included during model fitting to account for potential

across-year variability in landslide data reporting (sampling year; *YEAR*) and for the underlying repeated spatially nested data structure (sampling location; *LOC_ID*). These two random effects were averaged-out for the final prediction in analogy to Steger et al. (2021) (Section 4.3). The model output was visualized in the form of 2D surface plots and 3D perspective surface plots that depict the modelled probabilities as a function of the predicted variables (*T*, *P*, *DOY*). The resulting prediction surface plots were then complemented by binary thresholds (i.e., curves) that are directly related to ROC-based model performance

metrics (Section 4.4). Model validation was conducted using multiple CV procedures based on spatial partitioning of training and test sets (i.e., leaving specific subregions out), temporal data partitioning (i.e., leaving specific months or years out) and factor-based data partitioning (i.e., leaving environmental factors out). Finally, also a cross-check with recently occurred landslides data and data based on independent data sources was performed (Section 4.5).

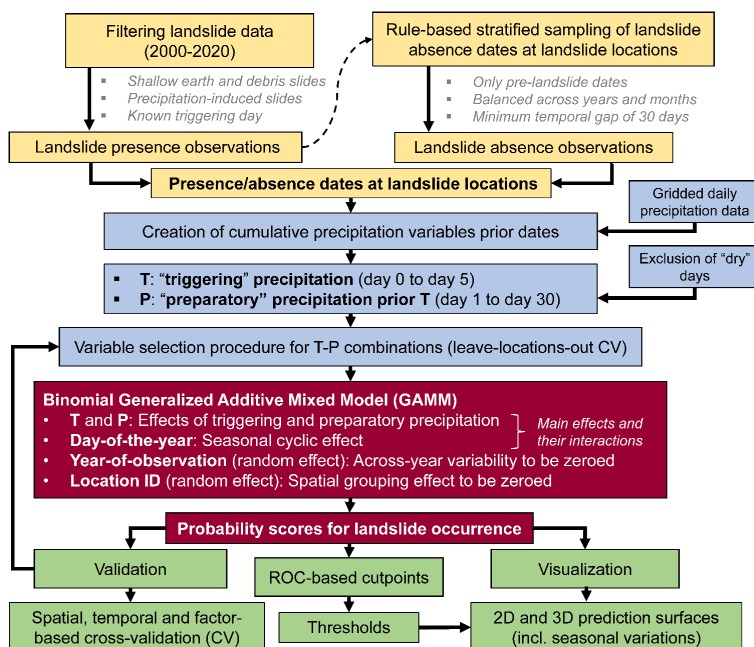


**Figure 3 Workflow of this study. The individual steps are described in detail in the Sections 4.1 to 4.5.**

### 4.1 Landslide data filtering and balanced sampling of landslide absence dates

Prior to the analyses, suitable landslide records were selected based on the filter criteria *"movement-type"*, *"material-type"*, *"movement-cause"* and *"date-availability"*. Only translational and rotational movement-types associated with the assigned

material-type *"earth"* and *"debris"* were included. In this context, we explicitly removed slide-types associated to deep-seated



gravitational slope deformations. Furthermore, only landslides with the assigned causes *"short-intense precipitation"* or *"prolonged precipitation"* were selected. The resulting 1822 entries were then filtered according to an additional temporal criterion: only entries with reliable information on the day of occurrence, and after the year 2000 were selected, resulting in a sample of 676 landslide records.

The 2021 IFFI data was still undergoing editing within the IdroGEO platform at the time of the filtering, and therefore the few available entries (n=3) were only used for the final cross-checks in analogy to the IRPI landslide data. All 47 IRPI landslide records were associated with time information that allowed to determine the required landslide occurrence day (exact time: 18, part of the day: 26, day only: 3). For validation, only non-fall type landslides with an assigned high spatial accuracy (i.e., mapping uncertainty $< 1$ km$^2$) were selected. Furthermore, entries with measured short-term precipitation lower than or equal

to 1.1 mm were also excluded (Section 4.2 for details). The final IRPI landslide data used for the cross-check consisted of seven shallow landslides and additional 12 slope instabilities associated with flow-type movements.

The outcome of a binary classification algorithm is heavily dependent on the ratio of landslide presence to absence across the variable space. For instance, using a single-variable classifier, a higher probability of landslide occurrence will be predicted for a specific variable characteristic (e.g., season: summer) in case the ratio of presences to absences is higher within this class

compared to the other values of this variable (e.g., season: winter). As a consequence, systematic biases can be introduced in a binary model by underrepresenting presence or absence information. This was demonstrated for spatial landslide susceptibility modelling in the context of systematically incomplete landslide data (Steger et al., 2017) and a spatially distorted absence sampling (Steger and Glade, 2017). We ensured that the previously described potential across-year reporting bias of landslide data is not propagated into the final predictions by introducing a dedicated random effect (Section 4.3).

The sampling of dates when landslides did not occur focused on drawing multiple absence observations prior to the known landslide date at each landslide location. This implies that the subsequent modelling relates to known landslide-prone terrain only, while areas with unknown occurrence of slope instabilities are disregarded. Rule-based stratified random sampling of absences was initiated by randomly selecting a high number of days (n=2000) between the year 2000 and 2020 for each known landslide position. Then, all absences related to a day after the landslide occurrence were excluded, since the effect of

precipitation on slope instability is likely to change after an event, especially when interventions took place. A minimum temporal distance of 30 days between the observations was introduced to avoid temporal overlaps of the cumulative precipitation time windows generated thereon and to reduce the effect of temporal autocorrelation amongst the observations. A final data-thinning step focused on balancing the observations across the years and months to ensure that each of these time periods is equally well represented by the absence data. For instance, the final absence sample is proportional to the number

of days within each month (e.g., January 31, February 28.25, April 30). Merging of the sampled absence data with the selected landslide presence observations resulted in the initial presence-absence sample with 6568 observations (676 presences, 5892 absences). Subsequent extraction of precipitation data for this initial sample allowed further exclusion of "dry" observations before modelling, as described in Section 4.2.



### 4.2 Cumulative precipitation variables T and P

The initial presence-absence sample (n=6568) built the basis for extracting pre-observation cumulative daily precipitation amounts for each sampled day. First, six day-windows (from day 0 to day 5) representing short-term cumulative daily precipitation before the observation day (*T*) were extracted. An observation day in the initial presence-absence sample refers to a 24-hour period from 00:00 UTC to 00:00 UTC. A day in the gridded precipitation data, instead, relates to a 24-hour period from 08:00 UTC of the previous day to 8:00 UTC of the observation day. Thus, during precipitation data extraction, a one-day

forward shift of the sampling date was required to ensure full coverage of precipitation amounts before the actual observation day. The drawback of this procedure is that precipitation potentially falling 8 hours after the observation day is also included in the *T* variables. The *T* day-window 1 (day 0 plus day 1) represents the shortest possible time window that fully covers the precipitation amount before the observation. It should be noted that the described one day-shift only concerned the described data extraction task, meaning that the original day assigned to each presence and absence observation remained untouched.

Second, several medium-term preparatory precipitation *P* variables were created by summing up the precipitation amounts that fell a multiple of days prior to *T* in analogy to Chleborad (2000). For this purpose, a one-day stepwise enlargement of the time-window for *P* was performed until the 30-day limit was reached. The resulting *T-P* combinations were then used to find the best pair of *T* and *P* to predict landslide occurrence (Section 4.3).

Third, data associated with a "dry" period shortly before or at the observation day was excluded from the analysis. A previous

landslide susceptibility study showed that the inclusion of a considerable portion of easy-to-classify "trivial" terrain, such as floodplains, can lead to an oversimplified classification task and problem-unspecific results (Steger and Glade, 2017). The same principle applies to this study when including a high portion of "trivial" time periods, namely observations that mainly refer to days without precipitation. Preliminary tests in this work confirmed that a binary model that included a high portion of "dry" time periods (i.e., non-precipitation observations) resulted in high predicted probabilities as soon as a small precipitation amount was recorded. To render our classification more problem-specific, we therefore excluded all days that did

not experience precipitation shortly before or at the observation date. For this purpose, we selected all observations where we had high confidence that they experienced some amount of short-term precipitation. In this context, the mean absolute error of the underlying precipitation grids (1.1 mm) was used as an exclusion criterion. The resulting final modelling sample (n=2832, 581 presences, 2251 absences) basically inherits the characteristics of the initial presence-absence sample while excluding

entries with measured precipitation equal or lower than 1.1 mm 1-day before or at the observation date.

### 4.3 Modelling using GAMMs and associated variable selection

Generalized Additive Mixed Models (GAMMs) have been recently used to assess landslide susceptibility in the context of incomplete landslide information (Steger et al., 2021; Lin et al., 2021). This study used a GAMM to discriminate precipitation days with landslides from precipitation days without landslides based on multiple variables. In summary, a GAMM *"allows*

*modeling of nonlinear functional relationships between covariates and outcomes where the shape of the function itself varies*


*between different grouping levels"* (Pedersen et al., 2019, page 1). Thus, it blends the strengths of a Generalized Additive Model (GAM), such as modelling nonlinear relationships, with advances of a Generalized Linear Mixed Model (GLMM), such as being able to account for between-group variability in the data, while simultaneously retaining a high model interpretability (Bolker et al., 2009; Zuur et al., 2009; Pedersen et al., 2019).

This work builds upon the comprehensive R package *mgcv* (Wood, 2004, 2011, 2017) that enables introducing different types of smoothing functions while offering an automatic smoothness estimation and a range of model evaluation and visualization options. The created model built upon four different types of smoothing functions, namely thin plate regression splines, cyclic cubic regression splines, tensor product interactions and random effects (Table 1).

Thin plate regression splines are general purpose splines and were used to model nonlinear relationships between the binary
outcome (landslide presence vs absence) and the precipitation variables $T$ (1st main effect) and $P$ (2nd main effect). Cyclic cubic regression splines are usually applied to model cyclic data in which the start and the end of the data series is matching. These splines were used to model seasonal effects based on the cyclic $DOY$ variable (3rd main effect).

The *mgcv* implementation also allows to model interactions between several smoothing functions by using tensor products. In cases the main effects are also present in the model, tensor product interaction terms are preferred to create an interpretable
model in presence of the main effects and any lower marginal interaction (Wood, 2017). Tensor product interactions were used to test the interacting effects of precipitation $T$ and $P$ on landslide occurrence (1st interaction term), the interacting effect of precipitation $T$ and $DOY$ (2nd interaction term) and the interacting effect of precipitation $P$ and $DOY$ (3rd interaction term).

In analogy to GLMMs, GAMMs also enable to handle hierarchical data structures by introducing random effects that account for variation between groups, such as data variability across temporal (e.g., years) or spatial units (e.g., sample locations). For
instance, if observational data is nested within sampling locations (e.g., multiple observations per location), a random intercept whose levels relate to the different sites can be introduced to capture variations deriving from the spatially nested data structure (Bolker et al., 2009; Pedersen et al., 2019). Random effects can also be used to capture variations related to an out-of-interest nuisance variable whose variability should be isolated to avoid confounding with other variables of interest (Steger et al., 2017). Two random intercept variables related to the sampling location and the sampling year were introduced. The categorical
variable $LOC\_ID$ (sampling location) accounts for the underlying nested data structure, i.e., the fact that the 2832 observations are spatially nested with the 581 sampling locations. The categorical variable $YEAR$ (sampling year) has been introduced to isolate a potential bias evolving from the underlying landslide data collection procedure in analogy to previous spatial landslide prediction studies (Steger et al., 2017, 2021; Loche et al., 2022). In detail, this $YEAR$ variable systematically captures data variability between the single years and therefore ensures that the modelled associations between our variables of interest ($T$,
$P$, $DOY$) and landslide occurrence are not systematically confounded by a potential inconsistent reporting of landslide occurrences across the year. It should be noted that both random effects were only used for model fitting to avoid confounded relationships between our variables of interest and landslide occurrence. For the applied model predictions and validation, these two terms were zeroed-out as described in Steger et al. (2017; 2021). This procedure allowed us to obtain model predictions and validation results that do not correspond to a specified year or location.






**Table 1 Model setup and variables introduced into the binomial GAMM.**

| Variable name(s) | Description | Smooth function / R command | Smooth term significance (p-values) | Effect used for prediction? |
|---|---|---|---|---|
| *SLIDEDATE* | Landslide occurrence dates (yes/no): Binary response related to presences (n=581) and absences (n=2251) | - | - | - |
| $T$ | Main effect of triggering precipitation: Short-term cumulative precipitation sum prior to and at the observation date | Thin plate regression spline / s($T$, bs="tp") | < 0.001 | Yes |
| $P$ | Main effect of preparatory precipitation: Medium-term cumulative precipitation sum prior to T | Thin plate regression spline / s($P$, bs="tp") | < 0.001 | Yes |
| *DOY* | Main seasonal effect on landslide occurrence described via the day-of-the-year | Cyclic cubic regression spline / s(*DOY*, bs="cc") | < 0.001 | Yes |
| $T * P$ | Interaction term that describes the interacting effects of T and P on landslide occurrence | Tensor product interaction / ti($T$, $P$, bs=c("tp", "tp") | 0.0124 | Yes |
| $T * DOY$ | Interaction term that describes the interacting effects of T and DOY on landslide occurrence | Tensor product interaction / ti($T$, *DOY*, bs=c("tp", "cc") | 0.0221 | Yes |
| $P * DOY$ | Interaction term that describes the interacting effects of P and DOY on landslide occurrence | Tensor product interaction / ti($P$, *DOY*, bs=c("tp", "cc") | 0.0108 | Yes |
| *YEAR* | Random intercept associated with the sampling year | Random effect / s(*YEAR*, bs="re") | < 0.001 | No (averaged-out via the *mgcv* exclude argument) |
| *LOC_ID* | Random intercept associated with the sampling location | Random effect / s(*LOC_ID*, bs="re") | 0.0884 | No (averaged-out via the *mgcv* exclude argument) |
| The model was fitted using the following command: fit = mgcv::gam(formula = *SLIDEDATE* ~ s($T$, bs="tp")+s($P$, bs="tp")+s(*DOY*, bs="cc")+ti($T$, $P$, bs=c("tp", "tp")+ti($T$, *DOY*, bs=c("tp", "cc")+ti($P$, *DOY*, bs=c("tp", "cc")+s(*YEAR*, bs="re")+s(*LOC_ID*, bs="re"), data=d, family=binomial, method="REML") | | | | |

The procedure to select optimal pre-observation cumulative precipitation time-windows for representing $T$ (n=6, day 0 to day

5) and $P$ (n=30; day 1 to 30) was based on the full model setup (see variable list in Table 1) and CV. In total, 165 $T$-$P$

combinations were tested using a two-parameter ($T_{days}$, $P_{days}$) grid-search combined with CV. To avoid temporal overlaps

between $T$ and $P$, 15 out of the possible 180 combinations had to be excluded (e.g., the $P_{days}$ search starts with the day 4 for

$T_{days}$ 3). Leave-locations-out CV was implemented by iteratively changing the respective $T_{days}$-$P_{days}$ combination. In this

context, 75% of randomly selected landslide locations, including associated pre-landslide absences at these locations, were

used for fitting the model while the remaining test set (25% of landslide locations) was used to validate the predictions using

the Area Under the ROC (AUROC). The AUROC metric can be seen as a threshold-independent measure of model

performance that depicts the degree to which a model is capable to discriminate presence observations from absence





observations (Hosmer et al., 2013). For each $T_{days}$-$P_{days}$ combination, this procedure was repeated 25 times and the median AUROC was taken as a final decision criterion. Computational feasibility of the 4125 model runs (165*25) was ensured by using the *mgcv*-function "*bam*" with fast REstricted Maximum Likelihood computation (fREML) (Wood, 2017). The best

performing *T-P* combination was then used to create the final GAMM with the default *mgcv*-function "*gam*" using REML for smoothing parameter estimation, since for this combination *bam* and *gam* results were almost identical.

**4.4 Visualization and thresholding**

The predictions of the final GAMM were first visualized by means of 2D surface plots and 3D perspective surface plots. The 2D surface plots show the predicted probability score between 0 and 1 for increasing values of two selected variables (e.g., *T*

on the y-axis and *DOY* on the x-axis). The perspective surface plots depict the predicted probabilities as a 3D surface in which the probability scores relate to the z-axis and two selected variables to the x- and y-axis. Since the model predictions are based on more than two variables, namely *T*, *P* and *DOY*, a strategy had to be found to deal with the not-shown third variable for visualization purposes. The not-shown remaining variable was either fixed to a specific value or averaged-out using the *mgcv-exclude* argument (cf. details in the figure captions). The modelled seasonal effect was first visualized separately by plotting

the predicted probability scores as a function of *DOY* and *T* and as a function of *DOY* and *P*. An intuitive visualization of the effect of short- and medium-term precipitation on landslide occurrence, further called *T-P* plots, was envisaged by plotting the predicted probability scores against increasing *T* (y-axis) and increasing *P* values (x-axis). Finally, a gradual increment of the day of the year (from 1 to 365) during prediction allowed the visualization of the seasonal effects of *T* and *P* on landslide occurrence via *DOY*-specific plots and a Graphics Interchange Format (GIF) animation.

Thresholds related to model performance metrics were added to the *T-P* plots by exploiting the ROC curve. The ROC plot visualizes the true positive rate (TPR; y-axis) against the inverse of the true negative rate (1-TNR = false positive rate; x-axis) for varying probability cutpoints. In our case, the TPR (also known as sensitivity, recall or hit rate) depicts for a given probability score the portion of correctly classified landslide observations (i.e., landslides exceeding the threshold) amongst all the observed landslide observations. The TNR focuses on the absence observations and shows for a given probability score

the portion of correctly classified "rainy" days without landslides (i.e., non-landslides below the threshold) amongst all observed days without landslides. A probability-threshold that relates to a high portion of correctly classified landslide presences (TPR) and a high portion of correctly classified landslide absences (TNR) is desired. A probability threshold associated to a TPR of 95% may be interpreted in analogy to a 5%-threshold that is frequently used for empirical rainfall thresholds, since it depicts that 95% of the observed landslides are exceeding the threshold while the remaining 5% are wrongly

classified (5% false-negative rate; FNR). In contrast to conventional empirical thresholds, the TNR can as well be extracted from the modelling results (i.e., probability space) giving valuable information on false alarms. For demonstration purposes, we extracted three probability thresholds. The first one can be considered an optimal cutpoint that refers to a probability score that is closest to the perfect classification in the ROC space (top-left corner). The remaining two thresholds refer to a very high





portion of correctly classified landslide observations (TPR 95%) and to a very high portion of correctly classified landslide
absence observations (TNR 95%).

### 4.5 Model validation

The model predictions were compared with the observations from numerous perspectives. The fitting performance of the model
was evaluated using the ROC and AUROC. The different resampling functions implemented in the R *sperrorest* package
(Brenning, 2012) were used to partition the data into training and test sets for the subsequent temporal, spatial and factor-based
CV. All these CV procedures are based on the principle of fitting the model with a subset of the data (training set) while then
testing the resultant predictions using the remaining independent test set via the AUROC metric. Temporal CV was conducted
by iteratively removing observations associated to one month (leave-one-month-out CV) or one year (leave-one-year-out CV)
from the training data sample and by testing the subsequent predictions on the left-out-observations using the AUROC. A
spatially explicit validation was achieved by leaving one-location-out (i.e., single locations are iteratively left out for testing)
and by leaving one-spatial-cluster-out (i.e., numerous adjacent locations) data partitioning strategies. The underlying spatial
clustering relies on the k-means method (Brenning, 2012) while we altered the average cluster size (k = 10, 25, 100) to obtain
validation results for single smaller (k = 100) and larger subregions (k = 10). Finally, CV was also performed to evaluate the
transferability of the modelling results across environmental units. In this context, leave-factor-out CV was based on
lithological zones (Fig. 2a), slope angle groups (Fig. 2b), elevation groups (Fig. 2c), geographical zones (Fig. 2d) and land
cover classes as described in Section 3.3.

### 5 Results

### 5.1 Sampling and variable selection

Landslide data selection and absence sampling led to an initial binary variable consisting of 6568 observation days (Fig. 4a).
This data relates to 676 sampling locations, with 676 landslide occurrence days (1 landslide per location) and 5892 pre-
landslide absence days. The grey bars in Fig. 4a highlight that the number of absences within each month is proportional to
the number of days of each month (i.e., each day-of-the-year was given an equal chance of being selected). This initial sample
represents landslide days and a balanced set of pre-landslide days without yet considering whether the respective days
experienced precipitation or not. The subsequent exclusion of "dry" observations led to the modelling sample shown in Fig.
4b. The exclusion of 3736 non-precipitation dates resulted in the final modelling sample consisting of 2832 observation days.
The comparison of Fig. 4a and Fig 4b shows that the precipitation filter mainly affected landslide absence observations with
more than 60% of all observations being excluded (from 5892 to 2251). In other words, our absence data mimics the yearly
distribution of "precipitation days" at landslide locations and suggests that less than 40% of the days experienced precipitation
shortly before or at the observation date (day 0 or day 1). The distribution of the filtered absence dates over the year (grey bars
in Fig. 4b) highlights that such "precipitation days" are more common during the summer months.

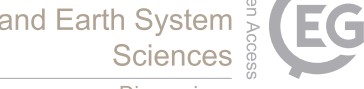

Despite having the label "precipitation-induced" in their attributes, 95 of the initial 676 landslide presence observations (14%) were removed from the precipitation filter. Our data therefore confirms that for 86% of these landslides (n=581) the respective location received some precipitation shortly before the landslide initiation. Comparing the seasonal distribution of sampled landslide presence days and absence days (blue vs grey bars in Fig. 4b) highlights that a relative high number of landslides can be observed for the months with the highest number of "precipitation days" (i.e., June, July, August). The highest number of

landslides per month was recorded in November, although this month has fewer "precipitation days". Without considering yet the associated amounts of precipitation, the data already suggests a seasonal effect, indicating that a rarer precipitation day in November is more likely to be associated with the occurrence of landslides than an average precipitation day during the summer.

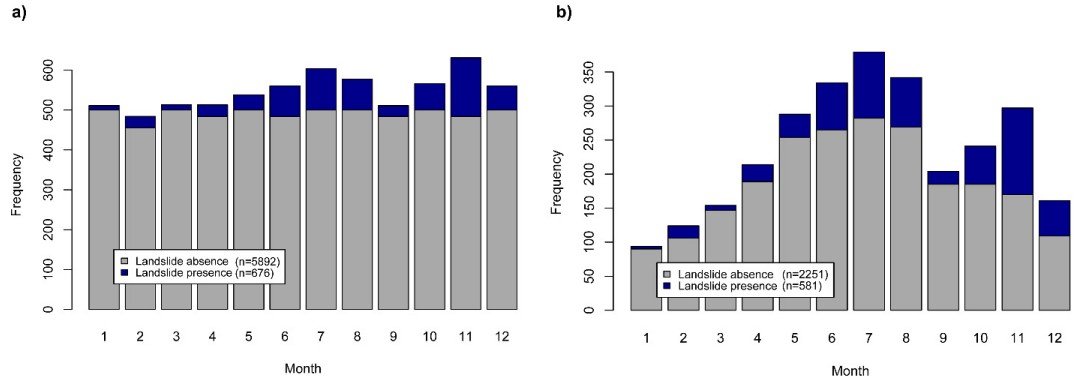

**Figure 4 Overview of the distribution of presence observations (selected IFFI data) and absences observations across the months. The initial presence-absence sample before excluding non-precipitation days (a) and the final modelling sample after excluding non-precipitation days (b). The distribution of grey bars in (a) reflects that each day of the year has an equal chance of being sampled, with the number of absence observations in each month being proportional to the number of days in each month. The grey bars in b) depict that the distribution of precipitation observations based thereon is considerably higher during summer.**

The final modelling sample (n=2832) was used in a repeated leave-locations-out CV framework to select the best performing time-windows for representing triggering precipitation ($T$) and medium-term preparatory precipitation ($P$) prior to $T$.

The pairwise confrontation of model performances is shown in Fig. 5. The highest performance (mean AUROC 0.87; median AUROC 0.86) was obtained by combining a 2-day cumulative precipitation variable ($T_{days}$ 1 representing day 0 plus day 1) with an antecedent 28-day time-window ($P_{days}$ 29). Considerably lower AUROCs were observed when combining a

comparably longer time-window for representing triggering precipitation (e.g., $T_{days}$ 3 to 5) with a relatively short time-window for representing preparatory precipitation. Lower performances were also observed for $T_{days}$ 0 that covers only 16 hours of the observation day and misses out on 8 hours of potentially crucial precipitation that are now represented within $T_{days}$ 1 (Section 4.2). In the following, the letters $T$ and $P$ will refer to the abovementioned 2-day and 28-day time-windows, respectively.



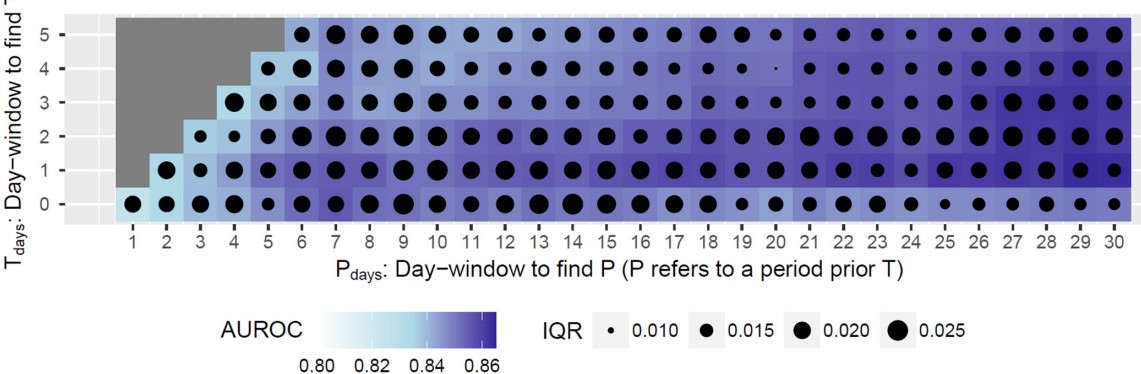

**Figure 5 Result of the pairwise selection of precipitation variables using leave-locations-out CV. The highest AUROCs (mean 0.87; median 0.86) were calculated when combining the day-window $T_{days}$ 1 with the day-window $P_{days}$ 29. $T_{days}$ 1 refers to the 2-day cumulative precipitation at day 0 plus day 1 while $P_{days}$ 29 refers to the 28-days prior $T$.**

The precipitation amounts associated with the selected $T$ and $P$ variables are shown in Fig. 6. The right skewed distribution of landslide absences observations for $T$ (Fig. 6a) depicts that the sampled precipitation days are mainly associated with low 2-day cumulative precipitation amounts of less than 10 mm. The median $T$ value for the absences relates to 6.9 mm (mean 11.6 mm) while the median for presences equals 33.6 mm (mean 42.4 mm). The 3$^{rd}$ quartile for $T$ equals 15.6 mm for the absences and 60.9 for the presences. For $T$ precipitation amounts >20 mm, a presence-to-absence ratio of ~1:1 (389 presences and 391 absences) can be found, despite the unbalanced initial ratio of presences to absences (1:3.9). Above 40 mm of $T$, three times more presences (n=246) than absences (n=82) are present.

Similar, but less pronounced tendencies can be observed for the medium-term precipitation variable $P$ with a generally increasing presence-to-absence ratios for higher amounts of $P$ precipitation. The median amount of $P$ precipitation for the absences equals 73.3 mm (mean 77.7 mm) and the respective median for the landslide occurrence dates equals 127.3 mm (mean 127.2 mm). The 3$^{rd}$ quartile of $P$ equals 105.2 mm for the absence observations and 163.3 mm for the presences. The ratio of presence-to-absence is balanced when considering $P$ precipitation >137 mm (252 observations in both groups).




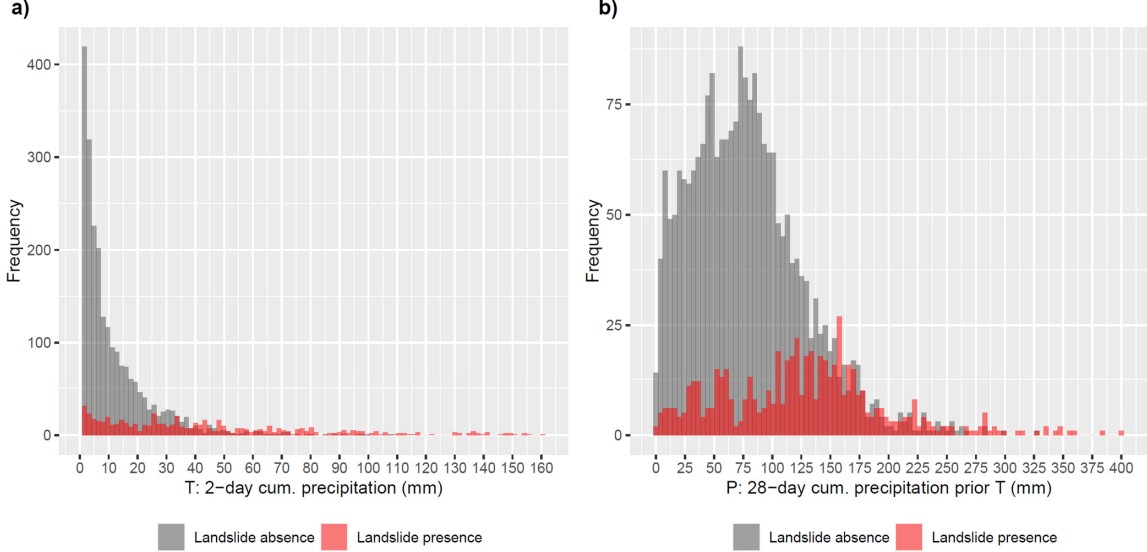

**Figure 6 Overlapping histograms for the presence (red) and absence (grey) observations for the two selected precipitation variables *T* (a) and *P* (b). Note that the ratio of presence-to-absence observations generally grows with increasing precipitation values indicating a trend of increasing landslide occurrence probabilities with increasing precipitation amounts.**

## 5.2 Model fit, modelled relationships and thresholds

The final model fit showed a high capability to discriminate landslide presence observations from absence observations with an AUROC of 0.87. All smooth terms used for prediction were significant with p-values < 0.001 for the main effects (*P*, *T*, *DOY*) and p-values < 0.05 for the three associated interaction terms (Table 1). The visualization of the predictions allowed insights into modelled relationships (Fig. 7). In general, increasing landslide probabilities were predicted for increasing amounts of *T* and *P* precipitation. The effect of short-term precipitation *T* on landslide occurrence was modelled to be rather constant over the year (e.g., horizontal contours in Fig. 7b) with the steepest increases in the predicted landslide probabilities for precipitation amounts between ~40 mm and ~70 mm (i.e., steep surface in Fig. 7a and narrow contours in Fig. 7b). Seasonal effects were observed to play a much more prominent role when considering our proxy variable for antecedent soil moisture conditions, namely the medium-term preparatory precipitation (*P*). For instance, the dip in the 3D surface (Fig. 7c) and the non-horizontal contours (Fig. 7d) highlight that the effect of *P* on landslide occurrence was modelled to vary systematically within a year. During winter (mid-February: day 40), a probability score of above 0.6 was predicted for a *P* amount of 150 mm (Fig. 7d). The same amount of preparatory precipitation, however, was associated with much lower landslide probabilities of around 0.2 during time periods characterized by dense vegetation and higher temperatures (day 240: end of August).


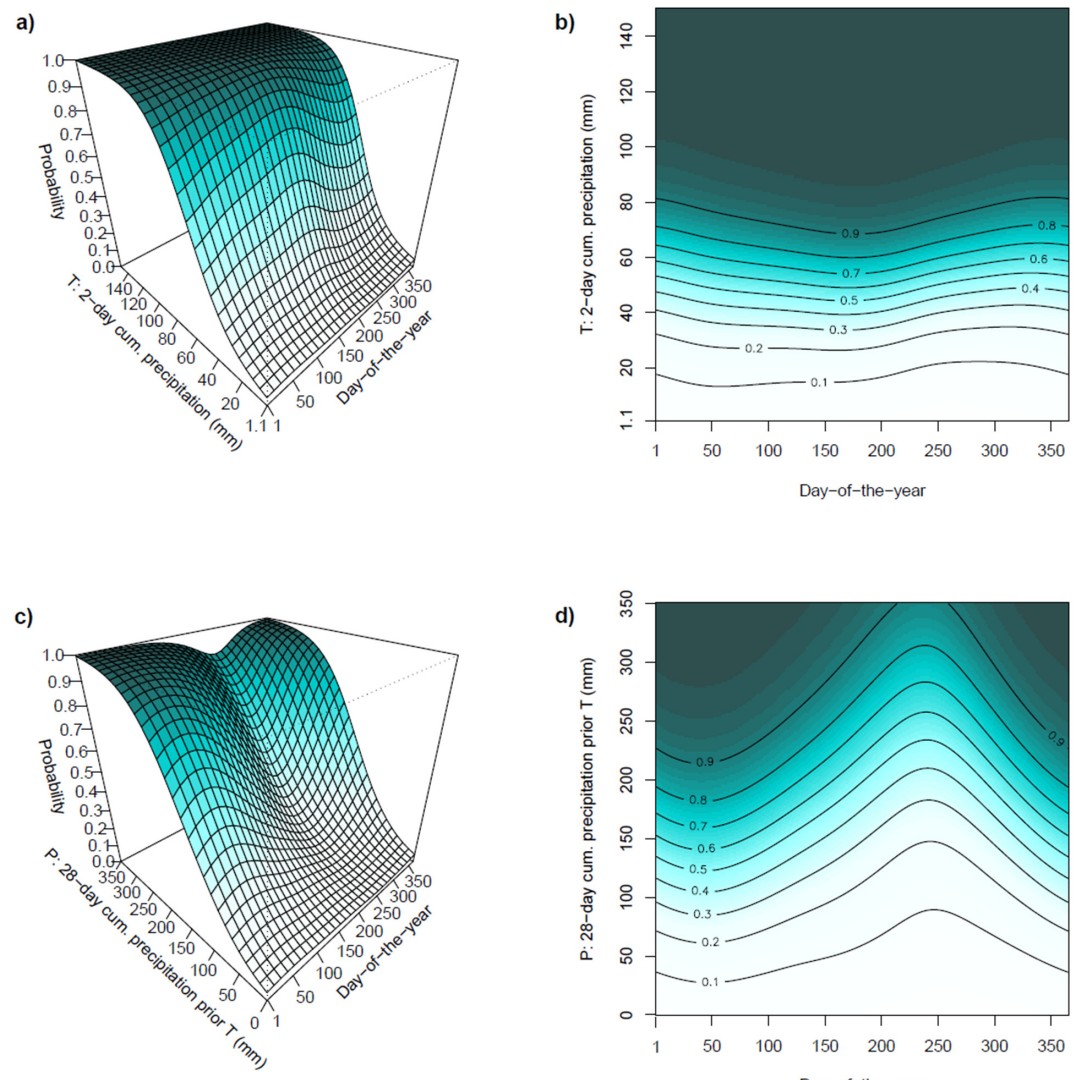

**Figure 7 The 3D surface plots (a, c) and 2D surface plots (b, d) depict the modelled effects of triggering precipitation *T* (a, b) and preparatory precipitation *P* (c, d) on landslide occurrence within a year (Day-of-the-year). The contours and the colour scale show predicted relative probability scores from 0 (white) to 1 (dark green). Note that these plots were created by fixing the not-shown variables (*P* in a, b; *T* in c, d) to their lowest value (*T* = 1.1; *P* = 0).**

The *T-P* plot (Fig. 8a) highlights the combined modelled effect of short-term precipitation *T* and the antecedent preparatory

precipitation *P* on landslide occurrence without yet considering seasonal variations (i.e., *DOY* was averaged-out for the



prediction). The highest probabilities were predicted for situations in which high *P* and high *T* occur simultaneously. The shown point positions depict the precipitation amounts for the underlying presence and absence observations. A comparably high portion of landslide observations (crosses) is observed for higher probability scores and a high amount of landslide absence observations (points) for lower probabilities. The ROC plot (Fig. 8b) refers to the final GAMM that includes seasonal

effects for the prediction. The plot depicts performance scores related to the three selected cutpoints. Since each of these cutpoints is associated with a probability score (first number in Fig. 8b), it can be drawn as a curve in the probability space (Fig. 8a).

The shown blue cutpoint refers to the a-priori selected TPR of 95%. It is shown that this cutpoint is associated with a low portion (40%) of correctly classified precipitation observations without landslide occurrence (TNR 40%). In contrast, the black

cutpoint refers to a particular high TNR of 0.95, but to a low portion of correctly classified landslide observations (TPR 56%). The "optimal" cutpoint in red balances misclassification rates and refers to a TPR of 81% and a TNR 79%. Note that these curves are drawn within Fig. 8a for demonstration purposes only to ease the understanding of the link between the ROC curve, the probability space, the observations, and the derived thresholds. A correct visualization of these thresholds is co-dependent on the third predicted variable, namely the seasonal effect (Fig. 9).

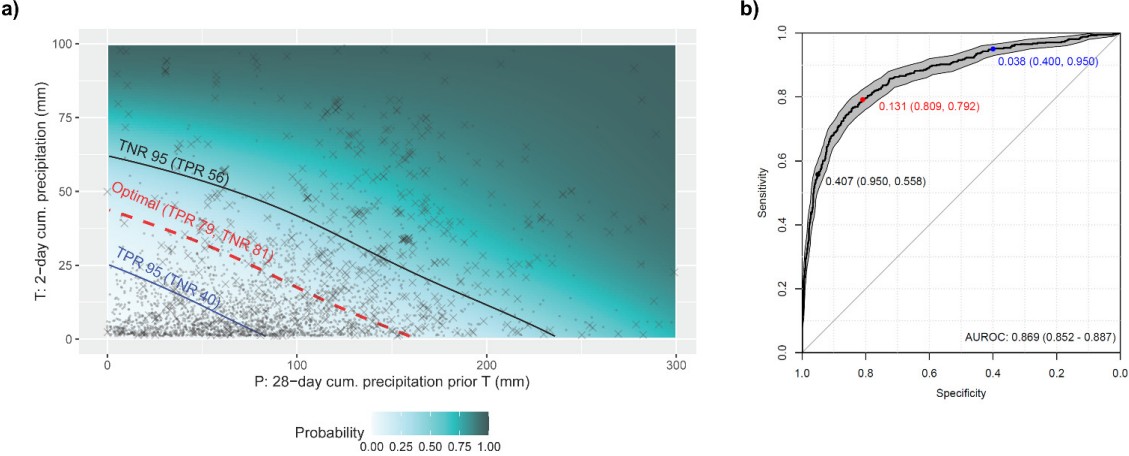


**Figure 8** *T-P* plot showing the predicted probability for increasing precipitation amounts (*T*, *P*) and associated thresholds (a). The thresholds are based on the ROC (b). The surface in (a) depicts the combined effect of short-term (*T*) and medium-term (*P*) precipitation on landslide occurrence for an average day-of-the-year (i.e., the *DOY* effect was zeroed for the prediction). The point positions relate to the precipitation amounts observed for the 581 landslide presence observations (crosses) and the 2251 absences

(points). Thresholds (a) can directly be derived from the ROC (b). The shown cutpoints refer to a probability threshold (first number) that is associated with a specific TPR (first number in brackets) and a TNR (second number in brackets). The optimal cutpoint is shown in red, the cutpoint related to a TNR of 95% is shown in black and the cutpoint related to a TPR of 95% is shown in blue.

The season-dependent thresholds in Fig. 9 reflect the previously described modelled relationships. For instance, comparing

the optimal thresholds across the seasons depicts that during winter times (Fig. 9a) a considerably lower amount of preparatory





precipitation is required to surpass the thresholds, compared to a day end of September (Fig. 9d). An examination of continuous time-series throughout the year ("Animation.gif" in the supplementary material) showed that critical landslide conditions were associated with the lowest amount of $P$ precipitation around mid-February (Day-of-the-year ~ 45). In contrast, the thresholds were associated with the highest amount of $P$ precipitation during early September (Day-of-the-year ~ 250). The comparatively

small seasonal variations in the modelled effect of $T$ on landslide occurrence are reflected by the modest changes in the threshold y-axis positions at low preparatory precipitation (e.g., at $P = 0$ mm).

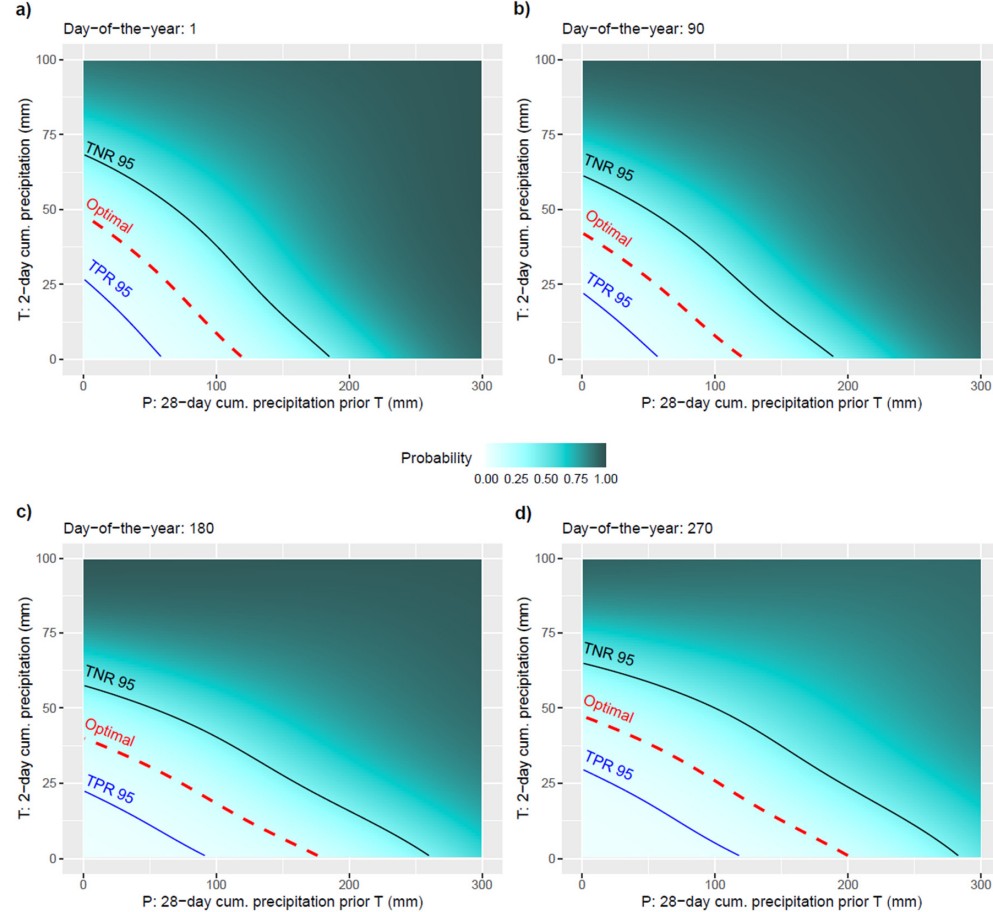

**Figure 9** ***T-P*** **plot showing the predicted probability for increasing precipitation amounts (*T*, *P*) and associated thresholds for a selected *DOY*: Day 1 (a), Day 90 (b), Day 180 (c) and Day 270 (d). The animation in the supplementary material (Animation.gif)**

**visualizes the continuous changes from day 1 to 365.**





### 5.4 Validation results

CV revealed the general robustness and transferability of the predictions with AUROC scores frequently exceeding 0.8. In general, a low performance for a specific unit (e.g., test subregion or test month) indicates that the observed conditions

responsible for landslide initiation within this unit were estimated to differ from the modelled conditions obtained by training a model within the remaining units. Temporal CV showed model transferability across the years (Fig. 10a) and months (Fig. 10b) with overall mean AUROCs of 0.83 and 0.84, respectively. A comparably high variation in model performance was observed when validating the model across the years with partly very high AUROCs of >0.95 for the years 2011, 2012, 2016 and 2018 and low AUROCs of <0.7 for the years 2001, 2006 and 2010. In this context also the partially low number of

underlying landslide observations used for testing must be mentioned. For instance, less than 10 landslides were inventoried for the years 2001 (n=6), 2003 (n=3), 2009 (n=9), 2010 (n=3), 2011 (n=1) and 2015 (n=5). More than 50 landslides were registered for the years 2007 (n=53), 2008 (n=106) and 2020 (n=85). The analysis of the measured precipitation amounts for the two years with the lowest performance scores (2001 and 2006) depicts that the associated landslide observations were related to particularly lower precipitation amounts, compared to the "mean landslide situation" in the data set. For instance,

the mean $T$ precipitation amount for all landslides was 42.4 mm while it was 10.1 mm and 16.9 mm for 2001 and 2006, respectively.

Leave-one-month-out CV led to more robust results with less variation in the estimated temporal transferability of the model (i.e., no AUROC below 0.7 or above 0.95). The results showed that the model performance is generally lower when predicting out-of-sample observations for spring (March, April, May) and June with AUROCs between 0.75 and 0.8. AUROCs above

0.85 were observed for the months January, February, July, August, October, November and December (Fig. 10b).
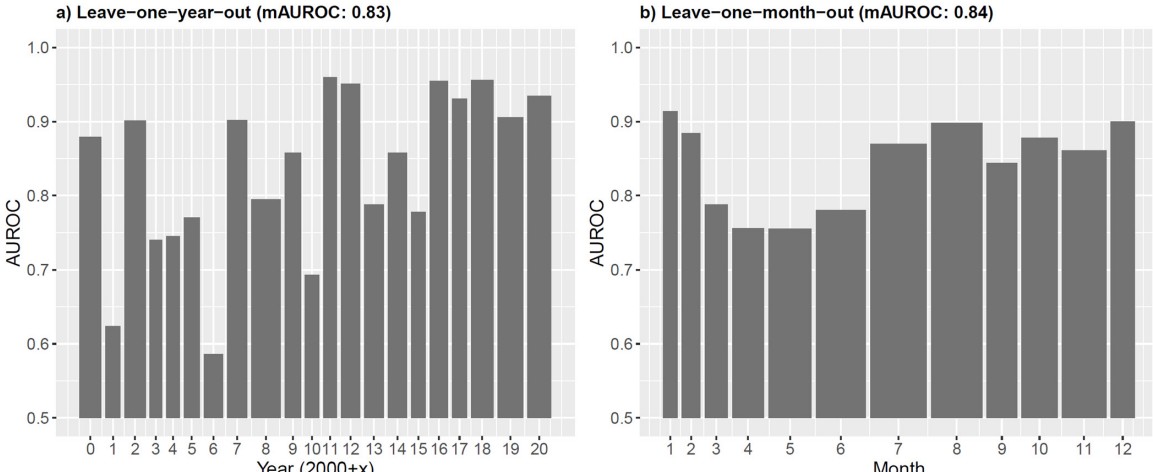

**Figure 10 Leave-one-temporal-unit-out CV. The AUROCs reveal, for each test year (a) or test month (b), the predictive performance**
**of a model that was trained on the remaining time periods (e.g., the months 1 to 11 were used to train a model whose prediction was**
**tested with month 12 data). The bar-width is proportional to the underlying test sample size (presences plus absences). The**
**mAUROC at the top refers to the mean AUROC of all time periods.**

Spatial CV led to mean AUROCs above 0.85 (Fig. 11). Leave-one-location-out CV showed an exceptionally high mean
AUROCs of 0.94 (Fig. 11a). These results revealed that 61.5% of all test locations were associated with a perfect AUROC of
1, meaning that the predicted probability score for the respective landslide observation was higher than the predicted
probability for any pre-landslide absence observation for the majority of test locations. Spatial CV using k-means clustering
(Fig. 11b – d) showed that observations within left-out regions can generally be well predicted by a model trained within other
subregions of South Tyrol. A more detailed elaboration of these results suggests a slightly lower model performances for the
Western part of the area. The influence of test sample size on variation in estimated performance scores is revealed by the
observed increasing standard deviation (SD) of AUROCs for an increasing number of data partitions (number of clusters k)
with a SD of 0.041 for k=10, a SD of 0.054 for k=25 and a SD of 0.119 for k=100.

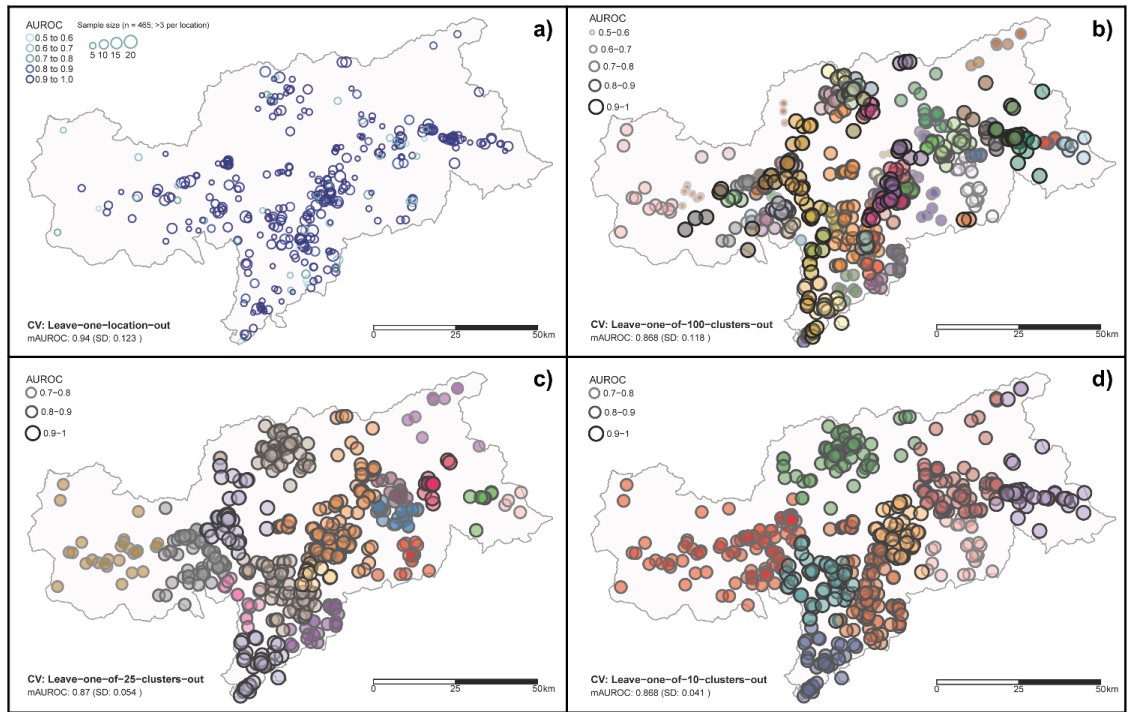

**Figure 11 Leave-one-spatial-unit-out CV. The maps depict, for different test subregions, the predictive performance of models that were trained on the remaining subregions. Leave-one-location-out CV (a) was performed for 465 locations associated with more than three observations per location. In a) The AUROCs are shown from light blue (0.5 – 0.6) to dark blue (0.9 – 1) and the circle size is proportional to the test sample size. The maps (b) to (d) are based on k-means clustering and depict different k-values (i.e., 100 clusters in b, 25 in c, 10 in d). Each colour is associated to a cluster while larger circles and darker circle borders are associated to higher AUROCs and vice versa. The AUROC at the bottom refers to the mean of all locations/clusters (mAUROC).**

Leave-one-factor-out CV showed mean AUROCs of >0.86 with no single unit being associated with an AUROCs lower than 0.75 (Fig. 12). AUROCs constantly above 0.85 depict that the modelled relationships are well transferable across the lithological units (Fig. 12a). Similar tendencies were observed when testing the model performance across the topographical variables slope and altitude (Fig. 12b and c). Slightly higher AUROCs were observed for models tested within less inclined terrain and at lower altitudes. This indicates that the precipitation conditions responsible for inducing a landslide are most different from the "average" situation for very steep terrain and at higher altitudes. In analogy, Figure 12d indicates that the performance of the model is lower when tested in the drier western part of South Tyrol (Fig. 12d) and when tested for forested terrain (Fig. 12e).


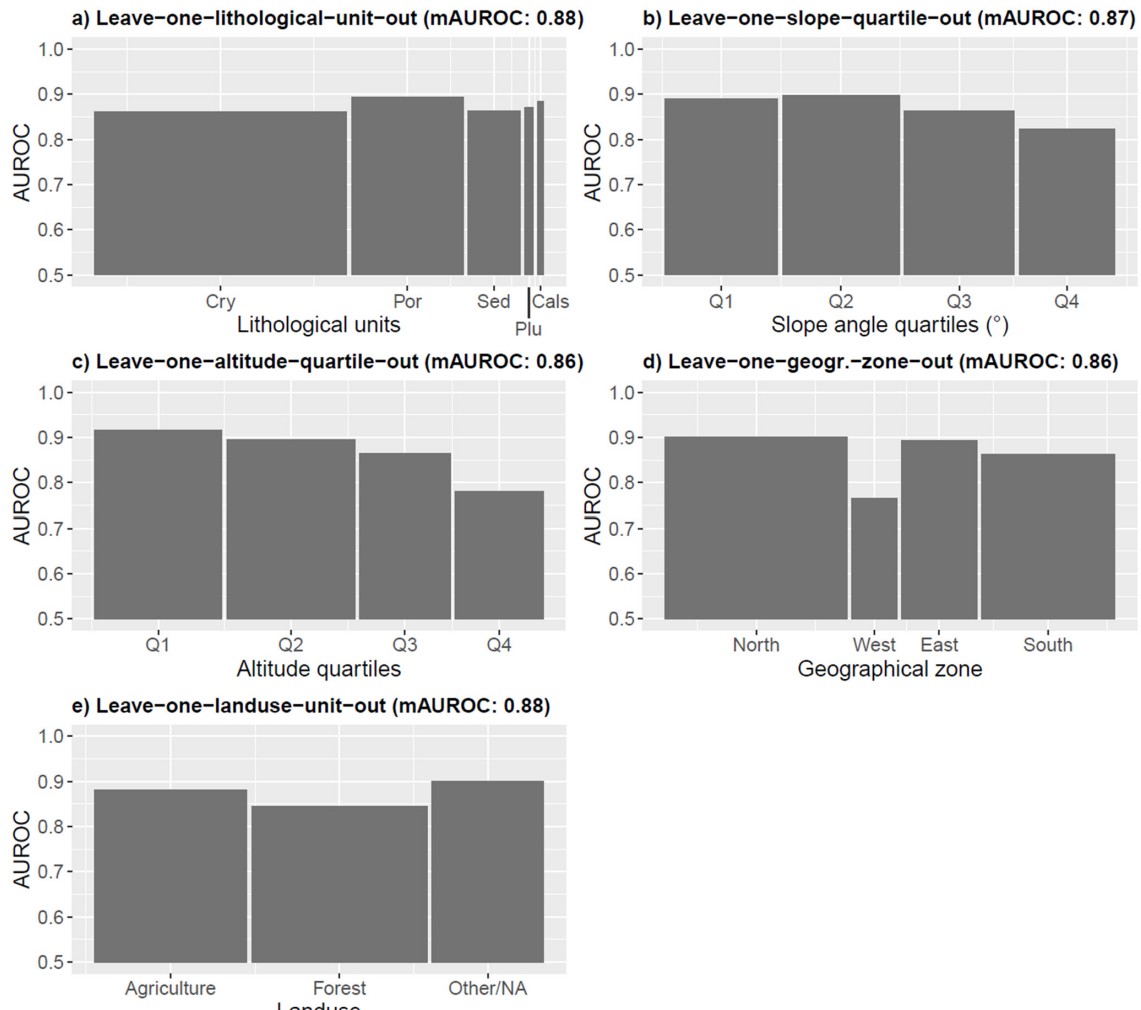


**Figure 12 Leave-one-factor-out CV. The AUROCs reveal, for each environmental test unit, the predictive performance of a model that was trained on the remaining environmental units (e.g., data associated to the slope quartiles Q1 to Q3 was used to train a model whose prediction was tested with slope quartile Q4 data). Bar-heights depict the test AUROCs for different lithological units (a), slope quartiles (b), altitude quartiles (c), geographical zones (d) and land use units (e). Lithological units: Crystalline basement (Cry),**

**Porphyry (Por), Sedimentary rocks (Sed), Plutonite (Plu) and Calcschists with ophiolites (Cals). The bar-width is proportional to**



**the underlying test sample size (presences plus absences). The mAUROC at the top refers to the mean AUROC across the shown units. Fig. 3 shows a map of the environmental units.**

Comparing the model predictions with the 2021 IFFI data and the independent IRPI landslide records generally confirmed a
high predictive performance of the model. The three IFFI landslide observations all occurred in August and were associated with a comparably high mean *T* precipitation amount of 56.6 mm and a high mean *P* precipitation amount of 238.6 mm, resulting in high predicted probabilities (> 0.78). Thus, all 2021 IFFI landslide entries exceeded the three thresholds (Fig. 9) by a considerable margin. Considering also pre-landslide absence observations at these locations led to an AUROC of 1, meaning that the respective landslide observations were associated with higher probabilities than any sampled pre-landslide
absence observations at the same locations. Lower but still high performances were computed on the basis of the independent IRPI landslide records. The AUROC for the 7 shallow landslide locations equalled 0.85 and 0.9 for the 12 precipitation-induced flow-type landslides. The 7 shallow landslide entries were associated with a mean *T* of 29.6 mm and a mean *P* of 98.1 mm resulting in a mean probability score of 0.31. Visually, six of these seven entries are located above the optimal threshold (Fig. 9). A mean *T* of 42.3 mm and a mean *P* of 129.3 mm were registered for the 12 flow-type landslides (i.e., 11 of the 12
entries were positioned above the optimal threshold).

## 6 Discussion

The seasonally dynamic predictive model allowed to combine the strengths of previously published approaches with the capability of novel GAMMs to identify season-dependent critical precipitation conditions for shallow landslide occurrences in the considered area. The model performed well, successfully accounting for the effect of short-term "triggering"
precipitation, medium-term "preparatory" precipitation, a cyclic seasonal effect and across-year data variation that potentially stems from an inconsistent landslide data reporting. Strengths, weaknesses and interpretation possibilities of the model are discussed below.

### 6.1 Antecedent precipitation and seasonal effects

Advantages of adding medium-term antecedent soil moisture and precipitation conditions have been highlighted by several
researchers (Crozier, 1999; Glade et al., 2000; Scheevel et al., 2017; Mirus et al., 2018; Monsieurs et al., 2019; Leonarduzzi and Molnar, 2020; Rosi et al., 2021; Stanley et al., 2021). In this study, medium-term antecedent precipitation conditions were found to have a statistically significant influence on the occurrence of shallow landslides in South Tyrol. The model provided quantitative evidence that higher amounts of preparatory precipitation (i.e., a proxy for antecedent soil moisture) decreases the necessary amount of short-term precipitation required to trigger a shallow slope instability (Crozier, 1989; Ponziani et al.,
2012). A major innovation of this research lies in the explicit and continuous modelling of seasonal variations of the effects of short- and medium-term precipitation on landslide occurrence. It was shown that during winter and spring a lower amount of preparatory precipitation is required to trigger a landslide. From a slope hydrology viewpoint, this trend could be expected,



since higher temperatures and a more active vegetation during summer and early autumn are likely associated with higher rates of evaporation, interception, transpiration and plant water absorption, all influencing soil water content and slope stability

(Crozier, 1989; Osman and Barakbah, 2006; Sidle and Ochiai, 2006; Norris, 2008; Gonzalez-Ollauri and Mickovski, 2017; Schmaltz et al., 2019). Such combined season-dependent effects of dynamic precipitation conditions on slope stability have not yet been considered in data-driven landslide prediction modelling. In a related recent work, Luna and Korup (2022) found that seasonal peaks in landslide activity are lagging annual precipitation peaks by one to two months, emphasising the effect of antecedent hillslope conditions. The underlying monthly models predicted, for a fixed monthly precipitation amount, a

considerably higher landslide activity for January/February compared to November/December when mean monthly precipitation amounts peaked.

With respect to the work of Luna and Korup (2022), our work shows similarities and differences in the way seasonal landslide patterns are reasoned and linked to potential drivers. Ultimately, it also reflects common challenges in model inference and causal interpretation. What a data-driven model enables to capture heavily depends on the underlying study design, which

includes the selection of variables used to capture potential confounding. Although we are aware that correlation does not necessarily imply geomorphic causation in data-driven landslide modelling (Steger et al., 2021), we suggest that the elaborated seasonal variation is primarily attributable to effects related to vegetation cover, and to a lower degree also to temperature and snow melting, rather than to delayed response of a hillslope to antecedent precipitation.

In detail, two landslide peaks were observed over the year in South Tyrol, one in July and one in November (Fig. 4). This trend

is likely to be reflected by a binary model trained with the cyclic *DOY* effect as its only predictor, since the input data showed highest presence-to-absence ratios within these periods. However, our final model additionally included two differently scaled precipitation variables capturing the important effects of short- and medium-term precipitation on landslide occurrence. Since these precipitation variables also exhibit systematic seasonal variation, they simultaneously captured a considerable portion of the seasonal variation in landslide occurrence. Thus, in the current model setup, only the "residual" seasonal variation that is

not explained by precipitation was explained by the seasonal *DOY* effect. In other words, this "residual" seasonal effect remained most influential in situations in which the included precipitation variables were less capable to explain seasonal alterations in landslide occurrence. The dip in the modelled seasonal effect (e.g., Fig. 7c), for instance, indicates that the precipitation variable *P* was less effective to discriminate landslide occurrences from non-occurrences during summer/early autumn (*DOY* ~ 200 to 250). Our interpretation that the slowly changing *DOY* effect mainly acts as a proxy for (unobserved)

vegetation/temperature influences, and not for antecedent precipitation amounts, already considers that potential time-lagged precipitation effects at a monthly scale are already accounted for by the 28-day precipitation variable *P*.

Seasonal effects were found to be most pronounced in spring (Fig. 7d at *DOY* ~ 50) when a less active vegetation cover temporally coincides with the main snowmelt period. These seasonal conditions are known to reduce slope stability (Crozier, 1989; Gonzalez-Ollauri and Mickovski, 2017; Krøgli et al., 2018). However, the lower model performance in spring (Fig. 10b)

suggests that the seasonal *DOY* effect could only partly compensate unobserved factors that (de)stabilize the hillslopes during this period. In this context, we assume that an inclusion of a variable that specifically describes snow melt conditions may



stimulate a further model improvement (Chleborad et al., 2008; Krøgli et al., 2018; Stanley et al., 2021). The modelled seasonal effect may also be influenced by variations in the precipitation regime and types across the seasons. In the north-eastern Alps, summer is the wettest season with prevailing convective activity, while in winter precipitation amounts are smoother and
mostly generated by larger-scale frontal events (Frei and Schär, 2001; Molnar and Burlando, 2008; Haslinger et al., 2019). Although a dedicated consideration of the particular precipitation regime/type would theoretically be possible, its practical feasibility might be hampered when considering a needed single-case assignment of precipitation characteristics for a high number of individual observations over 20 years (i.e., >2800 in this study). In summary, this section illustrated some of the challenges in deducing cause-and-effect relationships from data-driven landslide models while also emphasizing the
importance of controlling for confounding factors that may act at multiple scales (Steger et al., 2016, 2021). From a temporal perspective, our model captured effects related to four temporal scales, namely short-term ($T$), medium-term ($P$), seasonal ($DOY$) and across-year data variation ($YEAR$).

### 6.2 Mixed effects modelling and data sampling design

The explanatory power of the modelled relationships may be diminished not only by a non-consideration of confounding
factors, but also by the inclusion of unrepresentative input data. In binary modelling, systematic underrepresentation of presence or absence data is likely to provoke systematic model distortions, especially if such data bias is described by a model term (e.g., an underrepresentation of landslide data in winter can be reproduced by a seasonal effect). Within this study, mixed effects modelling allowed to isolate (and average-out) a potential inconsistent reporting of landslide observations across the years, as previously conducted for spatially inconsistent data in landslide susceptibility modelling (Steger et al., 2017, 2021).
The decision to sample absences only at the locations of inventoried first-time failures (i.e., sampling pre-landslide dates) substantially reduced the chance of precipitation being incorrectly flagged as an absence observation. It also implied that the created model systematically refers to landslide-prone terrain only.

The spatially nested sampling design (i.e., multiple observations at each location) resulted in a statistical dependence between the observations that must be considered. Conventional statistical models or machine learning techniques often ignore such
nested data structures. In this study, nesting of the data was explicitly addressed by using a location-specific random effect in a mixed-effects modelling framework (Bolker et al., 2009; Zuur et al., 2009). Temporal autocorrelation was counteracted by sampling only few observations at each location over time (five on average over 21 years) while additionally defining a minimum temporal interval of 30 days between the observations.

During absence sampling, each month was given an equal chance to be sampled as an absence observation, prior removing
non-precipitation observations (Fig. 4). Instead of simply sampling an equal number of "rainy" days for each month, we considered this two-step procedure necessary to achieve a representative within-year distribution of "rainy" days without landslides. The specific finding that an average "rainy" day in November is more critical than an average "rainy" day in summer would likely have remained hidden had the underlying sampling design not taken into consideration that "rainy" days are much more common in summer. In summary, these considerations highlight why a refined data sampling design, combined





with a specific strategy for handling potential bias in the input data, was considered important. Given the current trend of increasingly complex algorithms being applied to data-driven landslide prediction, we maintain that considerable effort should be placed on the underlying research design, input data quality, sampling strategy, correction of bias, model interpretability and a thorough validation of the results (Korup and Stolle, 2014; Goetz et al., 2015; Steger et al., 2016; Reichenbach et al., 2018; Steger et al., 2021; Tehrani et al., 2022). In this context, GAMMs can provide a valuable compromise between the

capabilities of "classical" statistical models (incl. mixed effects modelling) and highly flexible "black box" algorithms. For the temporal prediction of landslides, the application of very flexible algorithms, like deep learning, does not seem to be associated with a clear advantage so far (Tehrani et al., 2022).

### 6.3 Thresholds that relate to performance metrics

The presented model produced continuous outcomes (i.e., landslide probability scores) in contrast to conventional empirical

rainfall thresholds. Thus, it enabled to show that an increasing amount of precipitation was associated with a continuously increasing likelihood of landslide occurrence. However, the shown probability values are not straightforward to interpret, because they reflect the relative chance of landslide occurrence. These predicted values are heavily dependent on the underlying data distribution and the associated presence-to-absence ratio. For interpretation and practical applications, we therefore recommend linking such relative probability scores to more tangible model performance metrics, such as the

associated hit rate (TPR) or the false positive rate (1-TNR) (Fig. 8a). For instance, the meaning of a predicted probability of 0.13 (optimal threshold in Fig. 9) might be difficult to grasp and communicate to decision makers. Instead, the statement that 81% (TPR) of all landslide observations are above this threshold and 79% (TNR) of representative "rainy" days without landslides are below may facilitate understanding of the results. Linking predicted landslide probabilities to associated model performance metrics may also enhance practical applicability of landslide susceptibility maps that often rely on difficult-to-

interpret relative classifications (e.g., high, medium, low susceptibility) (Reichenbach et al., 2018).

Besides providing information on potential false alarm rates, an inclusion of non-landslide precipitation observations into the definition of critical rainfall conditions can additionally increase the robustness of the analysis and allow to derive optimized thresholds that balance misclassification rates (Frattini et al., 2009; Peres and Cancelliere, 2014; Gariano et al., 2015; Giannecchini et al., 2016; Postance et al., 2018; Leonarduzzi and Molnar, 2020). The threshold associated with the TPR of

95% (blue in Fig. 9) can be viewed as an analogue to the commonly shown empirical precipitation thresholds of 5%. Even though this threshold implies that a very high portion of the landslide observations were positioned above the threshold, the associated high false positive rate of 60% depicts that also the majority of "rainy" days without landslide occurrences exceeded the threshold. Relying on this 95% in decision-making would therefore come along with a high portion of false alarms. Thus, this study provides another illustration on why non-landslide events should be taken into account whenever continuous

precipitation records are available (Postance et al., 2018; Leonarduzzi and Molnar, 2020).



### 6.4 Multi-perspective model validation

A considerable amount of published research on critical landslide precipitation conditions does not include quantitative validation information (Gariano et al., 2015; Segoni et al., 2018a). Within this study, multiple data partitioning strategies combined with CV were used for in-depth model testing. In general, this procedure provided quantitative evidence of the

general model robustness and the model transferability across different time units (Fig. 10), across different areas (Fig. 11) and across different environmental conditions (Fig. 12). The additional consideration of "never-seen" landslide observations (i.e., 2021 IFFI data, IRPI landslide records) paired with constant plausibility checks of the modelled relationships provided a further confirmation of the model robustness.

Validation also acted as an additional analysis tool to identify situations in which the model performance deviates most from

the "average" condition. This in turn provided inspiration for further hypotheses to test and entry points for model improvement. For example, the systematically lower model performance on higher altitudes (Fig. 12c) and at steep terrain (Fig. 12b) indicates that in high alpine terrain and on steep slopes, landslide occurrence might be associated with different season-specific precipitation amounts, compared to lower lying and flatter areas. This in turn suggests that further model improvements may be achieved by including spatially explicit variables that directly describe landslide predisposition. The

current model was specified only for landslide-prone terrain and thus disregards important effects of spatially varying landslide predisposition by design (Crozier, 1989). Adding such spatial components within the current modelling framework may require an additional sampling of representative non-landslide locations within well-investigated areas and an exclusion of easy-to-classify "trivial" terrain in order to achieve problem-specific results (Steger et al., 2016; Steger and Glade, 2017; Bornaetxea et al., 2018; Knevels et al., 2020). This non-trivial step is planned within upcoming research.

The modelling framework presented in this study is flexible and can be adapted to model different types of sudden-onset processes that are determined by multiple interrelated dynamic variables acting on different time scales, such as debris flow or snow avalanche initiation. Novel non-linear mixed-effects modelling that incorporates additional group-specific interactions (e.g., smooth terms interacting with process groups) might also be valuable for multi-hazard assessment, i.e., for elaborating similarities and differences in the initiation conditions of compound sudden-onset hazards (Wood, 2017).

### 6 Conclusions

The developed approach using GAMMs enabled the identification of critical, season-dependent precipitation conditions for the occurrence of shallow landslides in the Italian province of South Tyrol. The flexible and interpretable model targeted a daily scale and considered precipitation that triggered shallow landslides as well as precipitation that did not induce slope instability. It simultaneously accounted for short-term and antecedent medium-term precipitation conditions while actively

counterbalancing a potential across-year landslide data reporting bias. Through our approach, we found that the same amount of preparatory precipitation led to different landslide probabilities depending on the day of the year. It was assumed that this seasonal effect was attributable mainly to temporal changes in vegetation and, to a lesser extent, to temperature and snowmelt.



The practical applicability of the results was enhanced by linking the predicted probability scores to common model performance metrics such as hit rates and false alarm rates, and by visualizing optimal and user-defined thresholds that change

dynamically throughout the year. The in-depth validation process confirmed a high predictive performance of the model and shed light on entry points for further model improvements. The developed approach represents a compromise between model transparency and model flexibility and can be adapted and extended to different study contexts. We believe that, despite the increasing availability of highly flexible data-driven algorithms, the quality of input data, study and sampling design, model transparency and result plausibility will continue to be very relevant for landslide prediction, especially if decisions are to be

made based on such results.

**Code and data availability**

Modelling has been conducted in R using the package mgcv. The R-code used to fit the model is shown in Table 1. Additional R-scripts and custom codes (e.g. data curation/preparation, model validation, thresholding, visualizations) are available upon

request. The landslide point data (unfiltered) can be accessed from https://idrogeo.isprambiente.it/app/page/open-data. The data sets of daily precipitation at 250 m resolution (NetCDF format) are freely available at PANGAEA Data Publisher for Earth and Environmental Science through https://doi.org/10.1594/PANGAEA.924502. Environmental data sets (lithological map, terrain model) can be accessed from the open geodatabase of the Autonomous Province of South Tyrol through http://geokatalog.buergernetz.bz.it/geokatalog/.


**Author contribution**

Steger: Conceptualization, Methodology, Analysis/Coding, Validation, Data curation, Visualization, Writing - Original Draft; Moreno: Conceptualization, Data preparation, Validation, Writing - Review & Editing; Crespi: Precipitation data generation, Validation, Writing - Review & Editing; Zellner: Data processing/management, Data curation, Coding, Writing - Review & 

Editing; Gariano, Brunetti, Melillo, Peruccacci, Marra: Conceptualization, Data provision, Validation, Writing - Review & Editing; Kohrs, Goetz: Conceptualization, Data pre-processing, Coding, Writing - Review & Editing; Mair: Conceptualization, Data provision, Validation; Pittore: Conceptualization,  Writing - Review & Editing.

**Competing interests**

The authors declare that they have no known competing financial interests or personal relationships that could have appeared to influence this work.

**Acknowledgments**

The research leading to these results are related to the Proslide project that received funding from the research program Research Südtirol/Alto Adige 2019 of the Autonomous Province of Bozen/Bolzano – Südtirol/Alto Adige. The authors are

grateful to the Autonomous Province of Bolzano for providing access to basic environmental input data. Many thanks also go



to Daniel Costantini and Silvia Tagnin of the Provincial Office for Geology and Building Materials Testing for their assistance in the preparation and selection of suitable landslide data.

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
