# Peer review of "Deciphering seasonal effects of triggering and preparatory precipitation for improved shallow landslide prediction using generalized additive mixed models"

_Natural Hazards and Earth System Sciences, 2022_

## Referee Comment (RC1)

**"General comments"**

The submitted paper presents an innovative approach to combine the effects of precipitation on shallow landslides at four temporal scales: triggering precipitation relevant at the short-term, preparatory precipitation acting at the medium-term, cyclic seasonal conditions and across-year variability (the latter being relevant to account for biases in reporting landslide data).

The analysis is based on using the generalized additive mixed models (GAMMs) to account for the interactions among the precipitation variables and successfully separate between precipitation conditions leading to failures and those not inducing landslide movements.

The produced outputs are landslide occurrence probabilities associated to the interplay between short-term triggering precipitation and antecedent preparatory precipitation in dependence on the seasonal period of the year.

The seasonal effect is interpreted as being mostly due to temporal changes in vegetation and secondarily to temperature and snowmelt.

The manuscript is well structured and written and illustrations and tables are all necessary. In my opinion, the paper is worth publishing with only some very minor/technical revisions.

**"Specific comments"**

A major strength and innovation of the study consists in the explicit consideration, the emphasis put on and the modeling of the seasonal imprints on the combined action of preparatory precipitation and short-term triggering precipitation in inducing slope failures at a daily scale.

The research is based on a data-driven modeling, applied following a thorough design of the study approach, and rendered highly transparently.

The robust development of the method, e.g. by eliminating the effects of year and locations, the careful and original design for sampling temporal landslide presence and absence observations at the mere landslide locations, and the data filtering criteria are well acknowledged.

The practical applicability of findings derives from the communication of predicted probability values also in terms of model performance metrics, such as *True Positive Rates* (TPR) or *True Negative Rates (TNR)*, as well as from the welcome modes for visualizing (e.g. 2D and 3D plots, animations) the resulted thresholds varying throughout the year. The value of the research also consists in the spatial

and temporal cross-validation of the results, which ensure a high robustness and adaptability of the model to other regions and temporal periods.

The paper is also to be appreciated for the R-scripts and codes made available as well as the open data.

**"Technical suggestions"**

- Section 3.1, page 6, line 160: if I understand correctly that the South Tyrolean data is part of the IFFI database, then I would suggest replacing "version" with "subset";
- Section 3.1, page 7, line 165: please delete the plural "s" from the word "landslides", since it should be the singular form;
- Section 3.1, page 7, lines 168-169: it is not clear what do you mean with "(incl. pre-2000 events)" in: " From January 2000 to the end of 2020, a total of 11420 points related to different movement types (incl. pre-2000 events) were registered for South Tyrol";
- Section 3.2, page 7, line 188: please replace "times" by "with": "multiplying the gridded daily anomalies with the gridded….";
- Section 3.3, page 8, lines 206-207: please adjust for the repeating word "based" and insert commas e.g.: "For the cross-validation (CV), based on a leave-one-factor-out data partitioning (Section 4.5), was based on several spatial environmental variables were used (Fig. 2).";
- Section 3.3, page 8, line 207: I would suggest replacing "this data" with "the data": "The data was obtained from the open Geodata platform of South Tyrol (Geokatalog, 2021).";
- Section 3.3, page 9, caption of Fig. 2: I would propose not using abbreviations in general in figure captions, thus I would propose replacing "CV" with "cross-validation";
- Section 4.3, page 12, line 313: please replace "This study used a GAMM to discriminate precipitation…" with "In this study, a GAMM was used to discriminate precipitation..";
- Section 4.3, page 13, lines 343-344: please replace "between" with "among" in: "In detail, this *YEAR* variable systematically captures data variability among the single years";
- Section 4.3, page 14, caption of Table 1: I would suggest adding in the caption information on the software used, e.g.: "Model setup and variables introduced into the binomial GAMM by means of/through the R software"; also, in the last row of the table, it would be clearer to specify "the following R command";
- Section 5.1, page 16: lines 413-424 present the resulted number of records in the landslide samples as well as the impact of filtering out "dry" observations; however, for a better understanding and avoidance of repetitions, referring to the corresponding methodological sections, also depicting the numbers of records issued after the various filtering phases, would be of help;
- Section 5.1, page 18, caption of Fig. 5: since illustrations have to be self-explanatory, please explain the abbreviation "IQR" used in the figure legend; I would also recommend not using the abbreviation CV in the caption but rather the full word;
- Section 5.4, page 24, caption of Fig. 10: similarly, I would recommend not using the abbreviation CV in the caption but rather the full word;
- Section 5.4, page 24, lines 555-556: please consider rephrasing as follows, using the plural: "…., meaning that, for the majority of the test locations, the predicted probability scores for the respective landslide observations were higher than the predicted probabilities for any pre-landslide absence observations."
- Section 5.4, page 24 line 558: please remove the word "a" from "suggests a slightly lower model performances" since you mean a plural term;

- Section 5.4, page 24 line 559: please insert "the" in two locations as follows:  "The influence of the test sample size on the variation in estimated…"
- Section 5.4, page 25, caption of Fig. 11: same observation, but  "CV" may be put into brackets after the first use of the entire word to explain the abbreviation used further in the caption as well as on the figures themselves;
- Section 5.4, page 25, caption of Fig. 11, line 565: please continue with lowercase letters after "In a)";
- Section 5.4, page 26, caption of Fig. 12: similar observation as for the latter figure captions, regarding the use of the abbreviation "CV";
- Section 5.4, page 27, caption of Fig. 12, line 588: I would suggest reversing the sentence into: "A map of the environmental units is shown in Fig. 3", since this puts the emphasis on the map to ease understanding;
- Section 5.4, page 27, lines 596-597: please consider reversing the phrase as follows and adding "respectively" in the end: "The AUROC equalled 0.85 for the 7 shallow landslide locations and 0.9 for the 12 precipitation-induced flow-type landslides, respectively.";
- Page 31, line 740: please change the number of this section title to "7 Conclusions"
- I would suggest being consistent throughout the paper when writing the name of the model ("generalized additive mixed models"), i.e. with either upper- or lowercase letters.

---

## Author Comment (AC1)

**Color scheme**

- Reviewer comments (RC): blue
- Answers to reviewer (i.e. author comments: AC): black

RC: "General comments"
The submitted paper presents an innovative approach to combine the effects of precipitation on shallow landslides at four temporal scales: triggering precipitation relevant at the short-term, preparatory precipitation acting at the medium-term, cyclic seasonal conditions and across-year variability (the latter being relevant to account for biases in reporting landslide data). The analysis is based on using the generalized additive mixed models (GAMMs) to account for the interactions among the precipitation variables and successfully separate between precipitation conditions leading to failures and those not inducing landslide movements. The produced outputs are landslide occurrence probabilities associated to the interplay between short-term triggering precipitation and antecedent preparatory precipitation in dependence on the seasonal period of the year. The seasonal effect is interpreted as being mostly due to temporal changes in vegetation and secondarily to temperature and snowmelt. The manuscript is well structured and written and illustrations and tables are all necessary. In my opinion, the paper is worth publishing with only some very minor/technical revisions.

AC: We thank Marta-Cristina Jurchescu very much for the thorough review of our manuscript. We are delighted that the work was judged to be "*worth publishing with only some very minor/technical revisions*".

RC: "Specific comments"

A major strength and innovation of the study consists in the explicit consideration, the emphasis put on and the modeling of the seasonal imprints on the combined action of preparatory precipitation and short-term triggering precipitation in inducing slope failures at a daily scale. The research is based on a data-driven modeling, applied following a thorough design of the study approach, and rendered highly transparently. The robust development of the method, e.g. by eliminating the effects of year and locations, the careful and original design for sampling temporal landslide presence and absence observations at the mere landslide locations, and the data filtering criteria are well acknowledged.

The practical applicability of findings derives from the communication of predicted probability values also in terms of model performance metrics, such as True Positive Rates (TPR) or True Negative Rates (TNR), as well as from the welcome modes for visualizing (e.g. 2D and 3D plots, animations) the resulted thresholds varying throughout the year. The value of the research also consists in the spatial and temporal cross-validation of the results, which ensure a high robustness and adaptability of the model to other regions and temporal periods. The paper is also to be appreciated for the R-scripts and codes made available as well as the open data.

AC: Thank you! We appreciate the positive feedback on our manuscript and are delighted to learn that the content of the paper, including technical details, has been well received by you.

RC: "Technical suggestions"

-      Section 3.1, page 6, line 160: if I understand correctly that the South Tyrolean data is part of the IFFI database, then I would suggest replacing "version" with "subset";

AC: Yes, this is correct. We will replace "version" with "subset".

RC: -      Section 3.1, page 7, line 165: please delete the plural "s" from the word "landslides", since it should be the singular form;

AC: This will be corrected.

RC: -      Section 3.1, page 7, lines 168-169: it is not clear what do you mean with "(incl. pre-2000 events)" in: " From January 2000 to the end of 2020, a total of 11420 points related to different movement types (incl. pre-2000 events) were registered for South Tyrol";

AC: Thank you for highlighting this issue. In the revised version, we will replace this sentence by *"For South Tyrol, the unfiltered IFFI data contains a total of 11420 points related to different movement types.".* The final sentence of this paragraph then points to the data filtering *"Prior to the analysis, both the IFFI and the IRPI landslide records were subjected to a comprehensive data filtering process as described in Section 4.1."*

In Section 4.1 we will then highlight details related to the temporal data filtering: *"…were then filtered according to an additional temporal criterion: only entries with reliable information on the day of occurrence, and from January 2000 to the end of 2020 were selected, resulting in a sample of 676 landslide records."*

RC: -      Section 3.2, page 7, line 188: please replace "times" by "with": "multiplying the gridded daily anomalies with the gridded…..";

AC: We will replace these terms as suggested.

RC: -      Section 3.3, page 8, lines 206-207: please adjust for the repeating word "based" and insert commas e.g.: "For the cross-validation (CV), based on a leave-one-factor-out data partitioning (Section 4.5), was based on several spatial environmental variables were used (Fig. 2).";

AC: We will replace the sentence with *"Cross-validation (CV) based on a leave-one-factor-out data partitioning (Section 4.5) focused on several spatial environmental variables (Fig. 2)."*

RC: -      Section 3.3, page 8, line 207: I would suggest replacing "this data" with "the data": "The data was obtained from the open Geodata platform of South Tyrol (Geokatalog, 2021).";

AC: We will replace "this data" with "the data" as suggested.

RC: - Section 3.3, page 9, caption of Fig. 2: I would propose not using abbreviations in general in figure captions, thus I would propose replacing "CV" with "cross-validation";

AC: We will modify the figure caption accordingly.

RC: - Section 4.3, page 12, line 313: please replace "This study used a GAMM to discriminate precipitation…" with "In this study, a GAMM was used to discriminate precipitation.."

AC: We will replace the sentence as suggested.

RC: - Section 4.3, page 13, lines 343-344: please replace "between" with "among" in: "In detail, this YEAR variable systematically captures data variability among the single years";

AC: We will rephrase this sentence by using the expression "*inter-annual*" as follows: "*In detail, this YEAR variable systematically captures inter-annual data variability (…)*".

RC: - Section 4.3, page 14, caption of Table 1: I would suggest adding in the caption information on the software used, e.g.: "Model setup and variables introduced into the binomial GAMM by means of/through the R software"; also, in the last row of the table, it would be clearer to specify "the following R command";

AC: We will modify the caption as follows *"Model setup and variables introduced into the binomial GAMM through the R software (package mgcv)."*

The last row of the table will be rephrased as suggested: *"The model was fitted using the following R command"*

RC: - Section 5.1, page 16: lines 413-424 present the resulted number of records in the landslide samples as well as the impact of filtering out "dry" observations; however, for a better understanding and avoidance of repetitions, referring to the corresponding methodological sections, also depicting the numbers of records issued after the various filtering phases, would be of help;

AC: Within the revised paper, we will start Section 5.1 with a description of the respective numbers as follows: *"The initial 11420 IFFI points were reduced to 2714 entries by first filtering translational and rotational slide-types and by excluding deep-seated movements. Most of these entries (n = 2319) were selected according to the subsequent material type filter, i.e. only "earth" or "debris" slides were considered. Further data subsampling according to the assigned movement cause "precipitation-induced" led to a subsample consisting of 1822 landslides. Out of these 1822 entries, 676 landslides were associated with reliable day information while occurring between January 2000 and the end of 2020."*

RC: - Section 5.1, page 18, caption of Fig. 5: since illustrations have to be self-explanatory, please explain the abbreviation "IQR" used in the figure legend; I would also recommend not using the abbreviation CV in the caption but rather the full word;

AC: We will replace "CV" with "cross-validation" and add *"Variability of AUROCs is shown by the interquartile range (IQR)."*

RC: -     Section 5.4, page 24, caption of Fig. 10: similarly, I would recommend not using the abbreviation CV in the caption but rather the full word;

AC: We will replace "CV" with "cross-validation".

RC: -     Section 5.4, page 24, lines 555-556: please consider rephrasing as follows, using the plural: "…., meaning that, for the majority of the test locations, the predicted probability scores for the respective landslide observations were higher than the predicted probabilities for any pre-landslide absence observations."

AC: Thank you for this suggestion. We will rephrase this sentence accordingly.

RC: -     Section 5.4, page 24 line 558: please remove the word "a" from "suggests a slightly lower model performances" since you mean a plural term;

AC: We will correct this.

RC: -     Section 5.4, page 24 line 559: please insert "the" in two locations as follows:  "The influence of the test sample size on the variation in estimated…"

AC: We will correct this as suggested.

RC: -     Section 5.4, page 25, caption of Fig. 11: same observation, but  "CV" may be put into brackets after the first use of the entire word to explain the abbreviation used further in the caption as well as on the figures themselves;

AC: We will modify this according to the suggestions.

RC: -     Section 5.4, page 25, caption of Fig. 11, line 565: please continue with lowercase letters after "In a)";

AC: We will correct this as suggested.

RC: -     Section 5.4, page 26, caption of Fig. 12: similar observation as for the latter figure captions, regarding the use of the abbreviation "CV";

AC: We will replace "CV" with "cross-validation".

RC: -     Section 5.4, page 27, caption of Fig. 12, line 588: I would suggest reversing the sentence into: "A map of the environmental units is shown in Fig. 3", since this puts the emphasis on the map to ease understanding;

AC: We will adapt the caption as suggested.

RC: -     Section 5.4, page 27, lines 596-597: please consider reversing the phrase as follows and adding "respectively" in the end: "The AUROC equalled 0.85 for the 7 shallow landslide locations and 0.9 for the 12 precipitation-induced flow-type landslides, respectively.";

AC: Thank you. We will reverse the phrase accordingly.

RC: -     Page 31, line 740: please change the number of this section title to "7 Conclusions"

AC: Thank you for highlighting this error. We will correct it accordingly.

RC: -     I would suggest being consistent throughout the paper when writing the name of the model ("generalized additive mixed models"), i.e. with either upper- or lowercase letters.

AC: We will check the entire paper and use the lowercase letters throughout the manuscript.

---

## Author Comment (AC2)

**Color scheme**
- Reviewer comments: blue
- Answers to reviewer (i.e. author comments: AC): black

RC: Dear Authors, dear Editor,

In the paper, the authors demonstrate a statistical data-driven model (generalised additive mixed model) for modelling landslide triggering conditions at a regional scale, i.e., in South Tyrol. The content of the paper is comprehensive, generally well written and interesting for the landslide community. However, before final acceptance, the manuscript needs to be improved by additional clarifications and revisions, which are listed below.

AC: We thank Reviewer #2 for the valuable feedback and the constructive suggestions. We are pleased that our manuscript was described as "*comprehensive, generally well written and interesting for the landslide community.*" Please find our point-to-point replies below.

RC: - Line 114: one of the goals of the paper is to identify critical seasonal precipitation conditions using GAMM, but throughout the paper these critical parameters are mentioned only very sporadically, so they are not apparent to the reader. I propose to create a table listing all critical parameters, both triggering and preparatory, along with thresholds and metrics from ROC.

AC: The reviewer is correct that the identification of critical season-dependent precipitation conditions for shallow landslides was one of the main goals of this research. The parameters used in the model are already highlighted within several parts of the methods section (i.e. from a more general description within Section 4 and Figure 3 to a more detailed technical elaboration within Section 4.2, Section 4.3 and Table 1).

As described in the results section (Line 491f), the highest landslide probability scores *"were predicted for situations in which high P and high T occur simultaneously"*. This continuous relationship is visualized within Figure 8a in the form of a prediction surface. Within our model setup, the amount of precipitation required to exceed a threshold is always dependent on a combination of three variables, namely short-term precipitation T, antecedent preparatory precipitation P and seasonality (DOY). This relationship is visualized within Figure 9 and the Animation GIF (supplementary material). From our viewpoint, the creation of a concise table showing the critical amount of precipitation required to induce a landslide is not feasible, particularly due to the non-trivial interplay of the three variables T, P and DOY (i.e. a large number of T-P-DOY combinations refer to a threshold). This is why we would prefer to use plots (i.e. probability spaces) and the Animation to visualize the elaborated critical conditions.

We agree that adding a table for the thresholds and metrics from ROC will further enhance transparency. Thus, we will add the following Table 2 to Section 5.2:

**Table 1  Probability thresholds and associated ROC-based metrics.**

| Threshold name | Probability | True positive rate (TPR) | True negative rate (TNR) |
|---|---|---|---|
| TPR95 | 0.04 | 95% (552 out of 581 presences) | 40% (900 out of 2251 absences) |
| Optimal | 0.13 | 79% (460 out of 581 presences) | 81% (1823 out of 2251 absences) |
| TNR95 | 0.41 | 56% (325 out of 581 presences) | 95% (2138 out of 2251 absences) |

RC: -Line 165: it is not enough to provide only a web link as a reference for the IdroGeo platform. Please add some paper reference(s) (e.g Idanza et al., 2021)

AC: Thank you for your suggestions. We will add the reference Iadanza et al., 2021 to the respective section in the manuscript:

Iadanza, C., Trigila, A., Starace, P., Dragoni, A., Biondo, T., and Roccisano, M.: IdroGEO: A Collaborative Web Mapping Application Based on REST API Services and Open Data on Landslides and Floods in Italy, ISPRS International Journal of Geo-Information, 10, 89, https://doi.org/10.3390/ijgi10020089, 2021.

RC: -Line 176: please explain which precipitation product is considered? It is clear that you have considered rainfall, but once you also mention snow and snowmelt, but it is not clear what exactly you have considered in the models under the term precipitation.

AC: As pointed out in the manuscript (Line 176f), the "*gridded fields of daily precipitation for South Tyrol were extracted from the 1980-2018 dataset produced by Crespi et al. (2021)*". This data was obtained by (Line 178f) *"(…) interpolating the rain-gauge daily records from a quality-checked and homogenized archive including around 80 station sites from the weather station network of South Tyrol*". Most rain gauges in South Tyrol are heated so that measured precipitation stems from both liquid and solid precipitation. This is why we opted to use the more general term "precipitation" instead of "rainfall" when describing our analyses and results within the paper. To enhance transparency, we will add the following sentence to this section:

"*Precipitation measurements are primarily related to rainfall, but in Winter, the amount of precipitation recorded may also include snowfall.*"

As mentioned in the discussion (Line 631f), the topic of snow melting directly refers to our interpretation of the seasonal effect, since we assume that "*the elaborated seasonal variation is primarily attributable to effects related to vegetation cover, and to a lower degree also to temperature and snow melting, rather than to delayed response of a hillslope to antecedent precipitation.*"

RC: -Line 243: missing reference to the criteria for landslides on the basis of which you made the selection

AC: The criteria used to select suitable "shallow landslide data" were not based on a specific publication, but defined specifically for the purpose of this analysis in collaboration with the entity that manages/collects the South Tyrolean landslide data (cf. acknowledgments: Provincial Office for Geology). The following description highlights the adopted criteria (Line 248f):

"*Prior to the analyses, suitable landslide records were selected based on the filter criteria "movement-type", "material-type", "movement-cause" and "date-availability". Only translational and rotational movement-types associated with the assigned material-type "earth" and "debris" were included. In this context, we explicitly removed slide-types associated to deep-seated gravitational slope deformations. Furthermore, only landslides with the assigned causes "short-intense precipitation" or "prolonged precipitation" were selected. The resulting 1822 entries were then filtered according to an additional temporal criterion: only entries with reliable information on the day of occurrence, and from January 2000 to the end of 2020 were selected, resulting in a sample of 676 landslide records.*"

Within the revised manuscript, we will add new text to the results section 5.1 to highlight the associated numbers (as requested by reviewer 1). This may further enhance traceability of the landslide data selection procedure.

"*The initial 11420 IFFI points were reduced to 2714 entries by first filtering translational and rotational slide-types and by excluding deep-seated movements. Most of these entries (n = 2319) were selected according to the subsequent material type filter, i.e. only "earth" or "debris" slides were considered. Further data subsampling according to the assigned movement cause "precipitation-induced" led to a subsample consisting of 1822 landslides. Out of these 1822 entries, 676 landslides were associated with reliable day information while occurring between January 2000 and the end of 2020.*"

RC: -Line 255-260: the number of training landslides for presence and absence days are unclear. Please improve this part with additional information and also edit Fig. 1a, where you have to distinguish training and validation landslides, for example with different colours.

AC: Within a previous NHESS publication, we elaborated advantages of k-fold cross-validation (CV) and k-fold spatial cross-validation (SCV) in comparison to conventional hold-out validation:

Steger, S., Brenning, A., Bell, R., Glade, T., 2016a. The propagation of inventory-based positional errors into statistical landslide susceptibility models. Nat. Hazards Earth Syst. Sci. 16, 2729–2745. https://doi.org/10.5194/nhess-16-2729-2016.

In this paper we highlighted that "*In contrast to single hold-out validation, CV and SCV are not based on one single split of the training and test sample (e.g. 80 % for calibration and 20 % for validation), but on a repeated partitioning of the original sample into k subsamples. In each iteration, a performance measure (e.g. AUROC) is estimated for one of the k subsamples, while*"

*the remaining (k−1) subsamples are combined into a training set that is used to calibrate the model. Thus, validation results that are based on CV and SCV are not dependent on one specific sample split.*" This is why "*CV as well as SCV allow that all available data can be used to validate and to calibrate the final models.*"

We therefore did not distinguish between training and validation landslides in our text or in the Figure, since each landslide is in fact both, a training landslide (within certain data partitions) and a validation landslide (within other data partitions). The ratio of training to test samples also varies between the different applied CV procedures (e.g. Leave-one-of-10-clusters out: 9:1; Leave-one-of-25-clusters out: 24:1; Leave-one-month-out: 11:1). However, we would prefer not to further expand on the description of the cross-validation procedure, as these techniques are established in the data-driven modeling community and we have already highlighted their benefits in the mentioned NHESS publication.

However, as described in the manuscript, 47 independent "IRPI landslide records" (cf. Section 3.1 and Section 4.1) were used for additional cross-checks. In this context, using the wording "cross-check" instead of "validation" was done on purpose, since we consider the comparison of our modelling results with landslide data from a very different data source of reduced explanatory power due to differences in e.g. the underlying definition of the process under investigation (i.e. shallow landslide) or differences in positional mapping uncertainty. Within the revised paper, we will add the following sentence to the description of the validation methods (Section 4.5):

"*Finally, the three independent IFFI entries from the year 2021 and the 47 IRPI landslide records were used to cross-check the model predictions.*"

RC: -Line 425: please explain why you removed 95 attributes

AC: We did not remove 95 attributes from the analyses, but 95 landslides. Within the paper we described that (Line 425f) "*Despite having the label "precipitation-induced" in their attributes, 95 of the initial 676 landslide presence observations (14%) were removed from the precipitation filter.*"

To further enhance transparency, we will rephrase this section to: "*After preliminary landslide data filtering, an additional 95 landslides were removed from the precipitation filter, because the available precipitation data provided no evidence that these recorded slope instabilities were primarily induced by precipitation.*"

RC: -Line 602: the discussion is too extensive; some parts are only informative and have nothing to do with the main goal of the paper, which is to decipher the seasonal effect of triggering and preparatory precipitation. I suggest to focus on the interpretation of the results, you could also refer to the new table in line 114 and focus on the critical parameters and their thresholds.

AC: We will shorten the discussion and remove discussion points that do not directly relate to the conducted research. We propose the following changes:

6.1 Antecedent precipitation and seasonal effects

This subchapter discusses season-dependent effects of dynamic precipitation conditions on slope stability, their potential causes while also highlighting challenges in causal model interpretation. We relate our work to the results of Luna and Korup 2022 that recently investigated seasonality in landslide occurrence using data-driven modelling. This discussion section directly relates to the main goal of the paper, which is why we would like to keep it as it is.

6.2      Mixed effects modelling and data sampling design

This subchapter discusses implications of input data quality issues (e.g. representativeness of landslide data) on the modelling results. It suggests methodical procedures that can be employed to achieve less biased results, which may be of particular interest to those in the scientific community seeking to replicate our approach.

In the interest of conciseness, we will remove the final section and associated references, which delves into broader methodological concerns, such as the choice between statistical models and more flexible "black box" algorithms. Furthermore, we will rename this section "Research design and representativeness of input data" to better convey its main focus.

6.3 Thresholds that relate to performance metrics

This section emphasizes the importance of creating thresholds in addition to continuous prediction surfaces to improve the practical applicability and interpretability of the results. In the revised version, we will eliminate the parts of the text that extend the discussion to landslide susceptibility studies to increase conciseness.

6.4 Multi-perspective model validation

This section focuses on the benefits of validating results from multiple perspectives. In line with this topic, we will remove sections that stray from this core topic. Specifically, we will delete the part that discusses the possibility of expanding the model into the spatial domain and the text that suggests the potential model's application in a multi-hazard context.

RC: You also need to better demonstrate the quality of the data, which you point out in the abstract but do not present and discuss enough in the paper.

AC: In the abstract we mention that "*the discussion illustrates why the quality of input data, study design and model transparency are crucial for landslide prediction using advanced data-driven techniques*". The revised discussion section 6.2 will now be fully dedicated to this topic, as described in the AC to the previous RC.

RC: - Line 750-755: in line with the introduction of this paper where you state: "reliable decision support tools" in this context you concluded that this approach needs further improvement. Please be more specific and give us a concrete evaluation of the applied approach as a promising tool for landslide early warning.

AC: In the revised manuscript, details on "further model improvements" will directly appear in the final discussion section and not in the conclusion (where we only mention that the validation process helped us to "*shed light on entry points for further model improvements*").

The final discussion section will read as follows: *"(…) This in turn provided inspiration for further hypotheses to test and entry points for model improvement. For example, the systematically lower model performance on higher altitudes (Fig. 12c) and at steep terrain (Fig. 12b) indicates that in high alpine terrain and on steep slopes, landslide occurrence might be associated with different season-specific precipitation amounts, compared to lower lying and flatter areas. This in turn suggests that further model improvements may be achieved by including spatially explicit variables that directly describe landslide predisposition. The current model was specified only for landslide-prone terrain and thus disregards effects of spatially varying landslide predisposition by design (Crozier, 1989)."*

In the revised paper (Section 6.3), we will add an example on how our model may be exploited for the purpose of early warning. The new text will read as follows:

*"The developed approach may be used in the context of early warning by exploiting the day-specific predictions, as shown in Figure 9. This involves evaluating whether observed or expected precipitation amounts are likely to exceed a particular threshold. To illustrate, consider a September day (Fig. 9d), where the optimal threshold will not be exceeded if 100 mm of precipitation were accumulated within the past 28 days (x-axis) and 25 mm of precipitation were expected to fall within the upcoming 2 days (y-axis). In contrast, the threshold will be exceeded if the same amount of short-term precipitation (i.e., 25 mm) falls after a particularly wet period (i.e., 200 mm within the past 28 days). Knowing that 81% of past landslides surpassed this threshold, while 79% of "wet" days without slope instability occurred below it, may further ease interpretation."*

---

## Author Response (AR1)

Dear Editor,

I am writing to submit the revised manuscript for our article entitled *"Deciphering seasonal effects of triggering and preparatory precipitation for improved shallow landslide prediction using generalized additive mixed models"*, which already underwent a rigorous peer review process.

We are pleased to inform you that we have carefully considered all of the comments and constructive criticisms provided by the reviewers, and have made the necessary revisions to our manuscript accordingly. The responses to all reviewer comments are already publicly available.

We hope that you will find the revised manuscript to be satisfactory for publication and appreciate the time and effort that the reviewers and editorial staff have dedicated to this manuscript.

We look forward to hearing from you soon.

Sincerely,

Stefan Steger and co-authors